# Circular single-stranded DNA as switchable vector for gene expression in mammalian cells

Linlin Tang[1,7], Zhijin Tian[2,3,7], Jin Cheng[1,7], Yijing Zhang[2,4], Yongxiu Song[2], Yan Liu[1], Jinghao Wang[2,3], Pengfei Zhang[2], Yonggang Ke [5] ✉, Friedrich C. Simmel [6] ✉ & Jie Song [1,2] ✉

Synthetic gene networks in mammalian cells are currently limited to either protein-based transcription factors or RNA-based regulators. Here, we demonstrate a regulatory approach based on circular single-stranded DNA (Css DNA), which can be used as an efficient expression vector with switchable activity, enabling gene regulation in mammalian cells. The Css DNA is transformed into its double-stranded form via DNA replication and used as vectors encoding a variety of different proteins in a wide range of cell lines as well as in mice. The rich repository of DNA nanotechnology allows to use sort single-stranded DNA effectors to fold Css DNA into DNA nanostructures of different complexity, leading the gene expression to programmable inhibition and subsequently re-activation via toehold-mediated strand displacement. The regulatory strategy from Css DNA can thus expand the molecular toolbox for the realization of synthetic regulatory networks with potential applications in genetic diagnosis and gene therapy.

Living cells have the ability to sense their environment, process internal and extracellular stimuli, and continuously respond to these inputs based on complex molecular programs. Synthetic biology aims to extend these programs with additional functions in order to realize designed cellular processes for useful purposes[1]. Inspired by electronic circuitry, the initial focus in synthetic biology was put on the implementation of engineered gene circuits, which were mostly implemented in bacterial hosts[2–4]. Regulators of gene expression are indispensable for the design of gene circuits, as they define the connections between the different circuit components and also enable sensing and responding to external signals. With the emergence of a wide range of tools for genetic engineering, implementations of synthetic biology in mammalian cells have recently made major progress[5,6].

In transcriptional regulation, natural transcription factors that are stimulated by specific inducer molecules to bind to or dissociate from DNA represent one of the most direct possibilities to induce or block the transcription process[7,8]. Besides, programmable transcription factors such as zinc-finger proteins (ZFP)[9], transcription activator-like effectors (TALE)[10,11], and CRISPR interference[12–15] have become popular tools for transcriptional regulation. At the post-transcriptional level, riboswitches and riboregulators have been widely used to control protein translation from switchable mRNA expression platforms via small molecule ligands[16], RNA molecules[17–19], or proteins[20]. Alternatively, the RNA interference (RNAi) machinery[21] has been utilized to control gene expression by modulating RNA stability.

The programmability of nucleic acid molecules bears an enormous application potential for the rational design of molecular devices

[1]Institute of Nano Biomedicine and Engineering, Department of Instrument Science and Engineering, School of Electronic Information and Electrical Engineering, Shanghai Jiao Tong University, 200240 Shanghai, China. [2]Hangzhou Institute of Medicine, Chinese Academy of Sciences, 310022 Hangzhou, Zhejiang, China. [3]Department of Chemistry, University of Science & Technology of China, 230026 Hefei, Anhui, China. [4]School of Life Sciences, Tianjin University, 300072 Tianjin, China. [5]Wallace H. Coulter Department of Biomedical Engineering, Georgia Institute of Technology and Emory University, Atlanta, GA 30322, USA. [6]Physics Department, Technische Universität München, Garching, Germany. [7]These authors contributed equally: Linlin Tang, Zhijin Tian, Jin Cheng. ✉e-mail: yonggang.ke@emory.edu; simmel@tum.de; sjie@sjtu.edu.cn

that operate in vivo. As biological DNA predominantly occurs in its double-stranded form, in vivo applications have so far mainly focused on RNA-DNA or RNA-RNA interactions, e.g., to regulate translation[17–19], RNA interference or CRISPR interference processes. Exploiting hybridization interactions between single-stranded DNA molecules and their associated conformational changes to perform useful functions in vivo has been rarely reported. One notable exception is the construction of topologically constrained DNA, which has been used to activate or repress transcription by the addition of specific DNA key strands in vitro and in bacteria[22].

In DNA nanotechnology, circular single-stranded DNA (Css DNA) are frequently used as a scaffold material for the folding of custom-shaped DNA origami objects with the assistance of short single-stranded "staple" strands[23,24]. Recent work[25,26] has reported the use of gene-encoding DNA origami structures for expression in mammalian cells, showing the great potential of such Css DNAs for applications in genetic engineering and synthetic biology.

Here we propose a strategy for the realization of reversible control of gene expression in mammalian cells, which is based on the use of Css DNA as a conformationally switchable genetic vector. We find that Css DNA containing the DNA sequence coding for enhanced green fluorescent protein (EGFP) can be efficiently transfected into mammalian cells via lipofection, and acts as a template for gene expression. Interestingly, addition of only a single, staple-like 48-nt-long blocking strand forming a closed loop in the Css DNA by pairing with two separate 24 nt sequence regions has a strong inhibitory effect on gene expression. Activation of gene expression from Css DNA clamped with locking strands can be achieved by supplying complementary trigger strands that remove the locking strands via toehold-mediated strand displacement. We constructed a Css DNA regulator consisting of a layered multi-input AND circuit, which can thus provide a sequence-programmable approach towards conditional gene expression in mammalian cells.

## Results

### Preparation and characterization of Css DNA by pScaf phagemid method

In order to conduct large-scale preparation of Css DNA molecules containing a protein-encoding sequence, we adopted the pScaf phagemid method[27], which was developed for the generation of arbitrary sequence scaffolds for DNA origami nanostructures. As shown in Fig. 1a, Supplementary Fig. 1, the pScaf phagemid method was used successfully to produce Css EGFP(+) DNA and Css EGFP(−) DNA (denoting the "sense" circular single-stranded EGFP in the 5′ to 3′ direction and "antisense" circular single-stranded EGFP in the 3′ to 5′ direction, sequences are given in Supplementary Tables 1 and 2), which contained the sense and antisense gene sequence segments, respectively, of two indispensable regions (a CMV promoter and the EGFP gene coding region) for the expression of an enhanced green fluorescence protein (EGFP). Agarose gel electrophoresis was first used to characterize the formation of Css EGFP(+) and Css EGFP(−). The clear, single gel band of Css EGFP(+) (2003 nt) indicated the high purity of the product (Fig. 1a). The AFM images of Css EGFP(+) in Fig. 1b showed that Css EGFP(+) adopts a rather compact conformation owing to the high flexibility of ssDNA[28], while plasmid DNA (pl DNA, i.e., circular double-stranded DNA) showed a more extended polymer contour, consistent with the much larger persistence length of dsDNA[29,30]. In order to demonstrate that the gene fragments of sense and antisense strand are indeed complementary (Supplementary Fig. 2), a hybridization experiment with Css EGFP(+) and Css EGFP(−) was successfully conducted, further confirming that Css EGFP(+) and Css EFGP(−) are correctly constructed and extracted. To demonstrate its single-stranded nature, we treated the Css EGFP(+) product with S1 nuclease, which specifically degrades ssDNA and leaves double-stranded DNA intact[31]. For Css EGFP(+) and P7560 (a single-stranded M13 viral

genome variant used as a positive control), the corresponding DNA bands disappeared after S1 nuclease treatment, while pl DNA (as a negative control) was only partially cleaved, but not degraded (Fig. 1c). Furthermore, as shown in Supplementary Fig. 3, we compared the stability of Css EGFP(+) and EGFP-mRNA (a linear single-stranded mRNA for EGFP expression), and found that Css EGFP(+) was more stable than EGFP-mRNA in DMEM medium supplemented with 10% FBS at 37 °C.

To confirm that Css DNA can be expressed in mammalian cells, we used the lipofectamine™ 2000 transfection reagent (lip2000) to deliver Css EGFP(+) and Css EGFP(−), respectively, into cultured MDCK cells, then detecting fluorescence after 24 h of culturing by flow cytometry. We used two observables to evaluate gene expression in our study: the fraction of cells showing fluorescence (termed transfection efficiency), and the mean fluorescence intensity per cell (a proxy for gene expression efficiency). The corresponding plasmid DNAs, termed pl EGFP(+) and pl EGFP(−), were used as a control. Consistently, as shown in Fig. 1d and Supplementary Fig. 4, an appreciable level of EGFP gene expression was achieved in MDCK cells transfected with Css EGFP(+) and Css EGFP(−), which had a high transfection efficiency (-95%; Fig. 1e).

### Verification for the replication of Css EGFP(+) and Css EGFP(−) in mammalian cells

During genome replication, double-stranded DNA genomes have to be unwound to serve as templates for semi-conservative replication, starting from RNA primers generated by dedicated RNA polymerases (primases). When single-stranded DNA is utilized as the genetic material – such as in single-stranded DNA viruses[32] and bacteriophages[33] – ssDNA is first converted into double-stranded form before it can be expressed. Based on our finding that both sense and antisense Css EGFP result in the expression of EGFP, we surmised that both Css EGFP(+) and Css EGFP(−) are transformed into double-stranded DNA by the cellular replication machinery before being expressed (Fig. 1f).

In order to verify this assumption, we transfected Css EGFP(−) into MDCK cells via lipofection, and extracted their whole DNA content (wDNA, which contains both endogenous genomic and foreign DNA) after culturing for 12 h. When the extracted wDNA is subjected to PCR amplification (Supplementary Table 7) in the presence of primer(+) (which pairs with the CMV region of Css EGFP(+), Fig. 1g, sequence is given in Supplementary Table 8), a product band clearly appears (Fig. 1h, Lane 2, encircled by the yellow dotted line) at the same position as the product obtained from PCR amplification of Css EGFP(+) prepared in vitro (Lane 1 in Fig. 1h). A similar result is obtained, when Css EGFP(+) is transfected into MDCK cells, and amplified after culturing using primer(−) (pairing with the EGFP region of Css EGFP(−), Fig. 1g, sequence is given in Supplementary Table 8) (Lane 4 and 5 in Fig. 1h). As a control, neither Css EGFP(+) nor Css EGFP(−) was transfected into MDCK cells, followed by PCR amplification of extracted DNA with primer(+) (Lane 3) or primer - (Lane 6). Based on the gel electrophoresis results, it can be concluded that both Css EGFP(+) and Css EGFP(−) likely have been converted into dsDNA inside the cells.

### Gene expression of Css DNA in various mammalians cell lines and in mice

To test the universality of Css DNA as a template for protein expression, we used lipofection to deliver Css EGFP and pl EGFP, respectively, into cells from 22 different mammalian cell lines such as MC-38, HELF, B16, etc. As shown in Fig. 2a and Supplementary Figs. 5–8, like pl EGFP, Css EGFP can be expressed in all these tested mammalian cells, and EGFP gene expression levels of Css DNA were similar to that of plasmid DNA. Notably, in some of the cell lines (such as HELF and WRL-68), Css DNA resulted in higher transfection efficiency than the plasmid. This might be related to their relatively lower molecular weight and higher

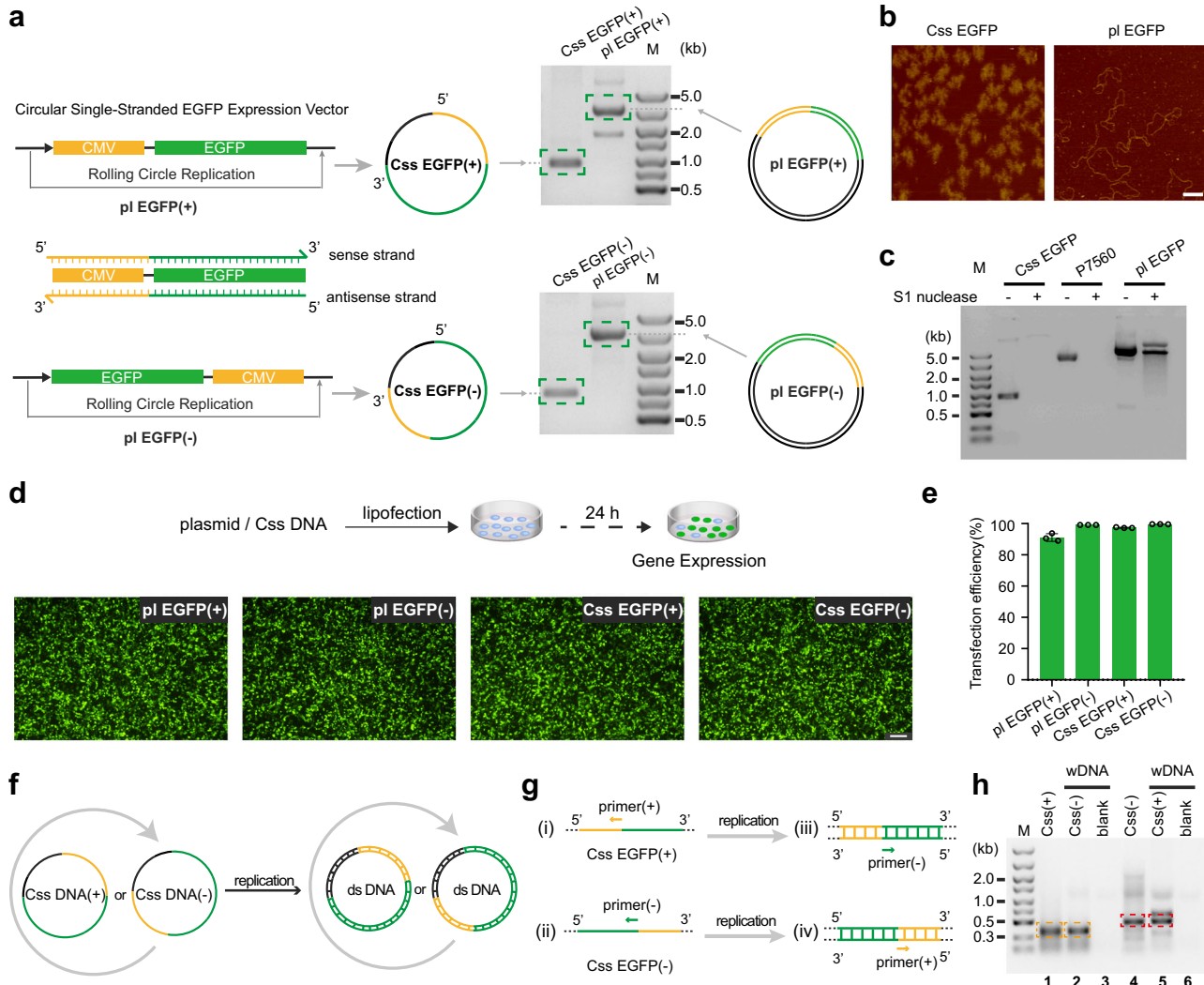

**Fig. 1 | Preparation and characterization of Css DNA for gene expression in mammalian cells. a** The production of sense circular single-stranded EGFP (Css EGFP(+)) and antisense circular single-stranded EGFP (Css EGFP (−)) via rolling circle replication on pScaf phagemid vector in Escherichia coli, and the 1% agarose gel analysis of Css EGFP(+), Css EGFP(−), pl EGFP(+) and pl EGFP(−). For Css DNA, CMV promoter sequence and EGFP coding sequence are shown in yellow and green, respectively, and the black region is a custom sequence with a fixed region of 393 bases that is required for the production of the Css DNA[27]. **b** AFM images analysis of Css EGFP and pl EGFP. Scale bar, 100 nm. **c** The 1% agarose gel analysis of Css EGFP, P7560 and pl EGFP (each 0.5 pmol) incubated with S1 nuclease (1 unit, Thermo Fisher) at 37 °C for 30 min. **d, e** Representative fluorescence images and transfection efficiency of MDCK cell line transfected with Css EGFP(+), Css EGFP(−), pl EGFP(+), and pl EGFP(−) (each 1.5 μg) after 24 h using liposomes. Scale bar,

100 μm. Data collected in **e** were quantified using flow cytometry and are presented as mean ± standard deviation (s.d.) for $n = 3$ biologically independent experiments, individual data points are overlaid, source data provided. **f** Illustration of replication of either Css EGFP(+) or Css EGFP(−) into double-stranded form. **g** Illustration of PCR amplification. The PCR amplification for (i) Css EGFP(+), (ii) Css EGFP(−), (iii) the extracted DNA from MDCK cells transfected with Css EGFP(+), (iv) the extracted DNA from MDCK cells transfected with Css EGFP(−). **h** Agarose gel (1.2%) analysis of PCR amplification products obtained using primer(+) with Css EGFP(+) (lane 1) or whole DNA (wDNA) extracted from MDCK cells transfected with Css EGFP(−) (lane 2), or using primer(−) with Css EGFP(−) (lane 4) or wDNA extracted form MDCK cells transfected with Css EGFP(+) (lane 5). In control groups, neither Css EGFP(+) nor Css EGFP(−) were transfected into MDCK cells, followed by PCR amplification of the wDNA with primer(+) (lane 3) or primer(−) (lane 6).

flexibility, and the abundant DNA replication machinery in these cells for the conversion of Css DNA to double-stranded form, eventually resulting in increased gene expression levels.

To address whether also other proteins can be expressed from Css DNA, we first constructed an additional Css mCherry(+) (2032 nt) encoding the red fluorescent protein mCherry, and a Css mCherry-EGFP(+) (3509 nt) simultaneously encoding the two fluorescent proteins mCherry and EGFP (Sequences are given in Supplementary Tables 3 and 4), which was achieved by inserting the corresponding gene sequences into the pScaf vector. As desired, gene expression occurred from both Css mCherry(+) and Css mCherry-EGFP(+)

(Fig. 2b, c). Satisfyingly, the cells transfected with Css mCherry-EGFP(+) were found to efficiently express both of the encoded fluorescence proteins. Both Css DNAs had similar transfection efficiencies (~75%) in cultured MDCK cells.

We additionally engineered a Css DNA construct that encodes the ALS (amyotrophic lateral sclerosis)-related protein FUS (fused in sarcoma), which was tagged with a red fluorescent protein (RFP). Using lipofection, the Css RFP-FUS(+) (4751 nt, 1 μg) (Sequences are given in Supplementary Table 5) was introduced into HeLa cells, followed by cell culture for a period of 36 h to allow for the expression of the construct. As an RNA-binding protein, FUS is known to display liquid–liquid phase

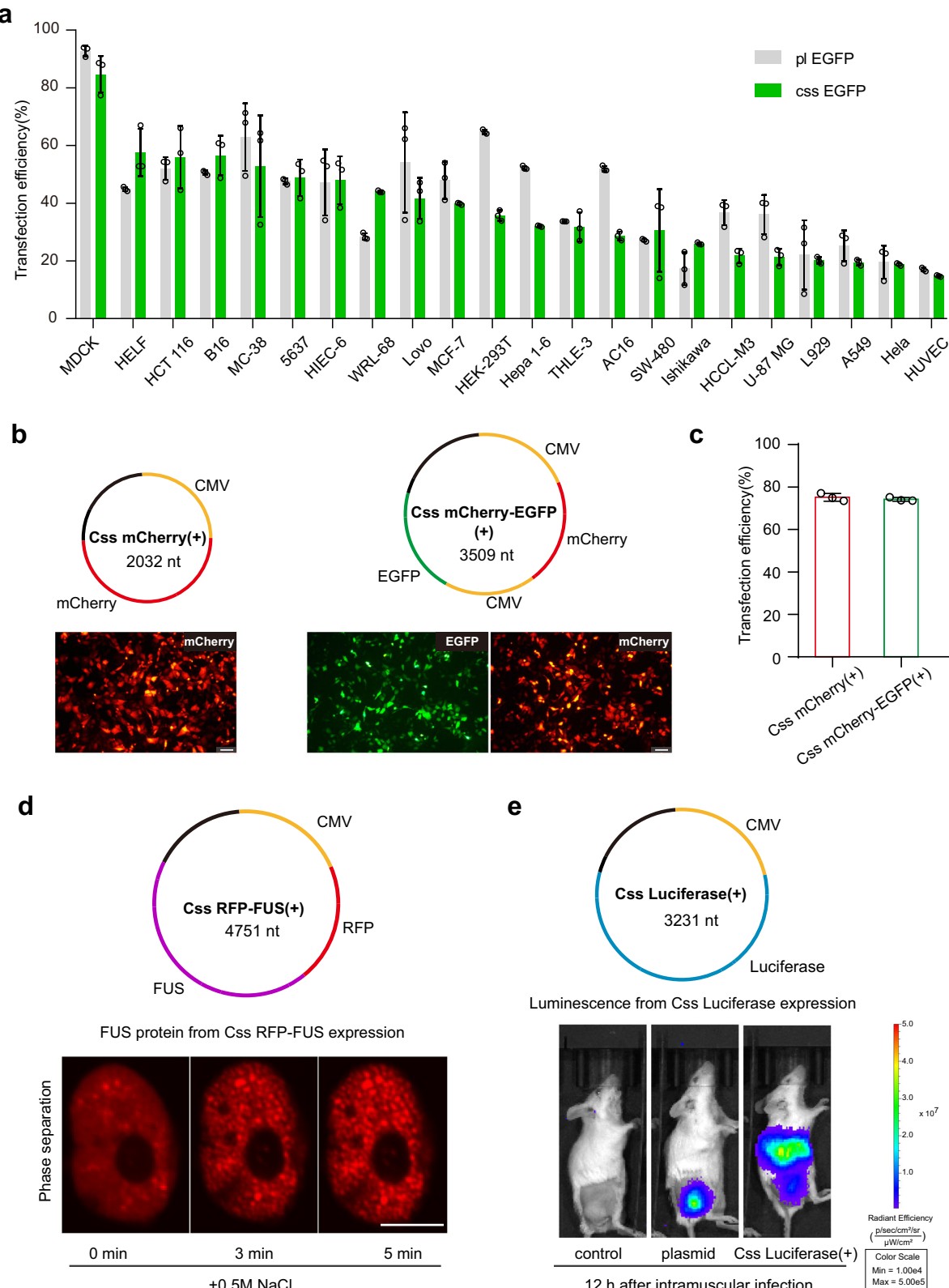

separation in the nucleus. When exposing transfected HeLa cells to hyperosmotic stress by the addition of 0.5 M NaCl to the medium for 0 min, 3 min, and 5 min, dynamic phase-separation behavior involving the assembly of nuclear RFP-FUS granules could be monitored using confocal microscopy (Fig. 2d). This observation was consistent with previously reported results[34], demonstrating that Css RFP-FUS(+) indeed successfully led to the expression of FUS protein in the Hela cells.

To further test the potential of Css DNA for in vivo protein expression, we constructed Css Luciferase(+) (3231 nt, Sequences are given in Supplementary Table 6) encoding a luciferase protein that generates luminescence in the presence of luciferin substrates. In vivo luminescence images of mice were captured 12 h post-intramuscular (i.m.) injection with lipid nanoparticles (LNP) loaded with the Css Luciferase. An experimental group receiving physiological saline served

**Fig. 2 | Gene expression of Css DNA in mammalian cells and in mice. a** The expression efficiency of various mammalian cells transfected with Css EGFP and pl EGFP (each 0.5 pmol), respectively, via lipofection. Css EGFP was obtained from pl EGFP via enzyme digestion. Error bars represent standard deviations from at least three independent tests. **b, c** Representative fluorescence microscopy images and transfection efficiency of MDCK cell line transfected with Css mCherry(+) and Css mCherry-EGFP(+) (each 0.25 pmol), respectively, after 24 h using liposomes. Scale bar, 50 μm for all fluorescence microscopy images. **d** Representative images of Hela cells assembled into nuclear RFP-FUS granules induced by 0.5 M NaCl dissolved in the complete medium (DCM) for 0 min, 3 min, and 5 min, respectively. In **b** and **d**, the images are representative of one of *n* = 3 biologically independent experiments; similar results were observed each time. Scale bar, 10 μm. **e** In vivo

luminescence images of mice collected 12 h after intramuscular administration of physiological saline (control), LNP-plasmid and LNP-Css Luciferase, respectively. The LNPs were injected into the right thigh muscle. The Css Luciferase was mainly expressed in liver of mice, but the plasmid Luciferase was mainly expressed at the injection site. The images are representative of one of *n* = 2 biologically independent experiments; similar results were observed each time. Data collected in **a** and **c** were quantified using flow cytometry and are presented as mean ± standard deviation (s.d.) for *n* = 3 biologically independent experiments, individual data points are overlaid, source data provided. For schematic of Css DNA in **b, d** and **e**, CMV promoter sequence, EGFP coding sequence, mCherry coding sequence, FUS coding sequence and Luciferase coding sequence were shown in yellow, green, red, purple, and blue, respectively.

as a negative control, while a group receiving LNPs carrying the plasmid for luciferase served as a positive control for luciferase expression. As depicted in Fig. 2e and Supplementary Fig. 9, Css DNA encoding luciferase results in an appreciable expression in liver of mice, and even had a higher protein expression level 12 h after injection compared to the positive control group (although the plasmid was expressed mainly at the injection site). We surmise that some of the LNPs would enter the liver for both Css DNA and plasmid DNA. However, since the plasmid Luciferase (5984 bp) has the bacterial backbone sequence (required for the production of plasmid DNA), such as an antibiotic resistance gene, an origin of replication, etc., the plasmid may be more easily degraded and cleared by the liver, leading to comparatively stronger luciferase expression from Css DNA within 12 h. As shown in Supplementary Fig. 9, however, it is found that with increasing time from 0.5 to 47 days, the expressed protein in liver would be metabolically cleared, and both gene expression of Css DNA and plasmids then mainly occurred at the injection site. Taken together, these results demonstrate that Css DNA can be utilized as an efficient gene expression vector for use in mammalian cells and in mice.

**Exploration of the effect of short complementary strands on Css DNA expression**

Considering the replication mechanism from Css DNA into dsDNA, we tested whether complementary DNA strands on Css DNA molecule would have an effect on gene expression from Css DNA. The enzyme digestion method[35,36], with nicking enzyme Nb.BbvCI and Exonuclease III (ExoIII) to digest one nicked single-stranded DNA directly from the plasmid (pl) DNA, was employed to obtain Css DNA and hybrid Css-dsDNA. A plasmid DNA (total gene length of 4732 bp) was coded with a EGFP sequence for later gene expression. As shown in Supplementary Fig. 10, we characterized the time-dependent conformational changes of pl DNA with a nicked strand upon digestion with ExoIII at 28 °C by agarose gel (1%) and atomic force microscope (AFM). The initially supercoiled pl EGFP (gel band V) is shown to relax into a nicked circular double-stranded form with a lower band mobility after nicking by Nb.BbvCI (pl EGFP- Nb.BbvCI, gel band V_Nb). The increase in gel band mobility of the other products with digestion time (IV-I; 5 min, 10 min, 20 min, and 30 min respectively) is also shown in Supplementary Fig. 10a, b. We further calculated the contour length (from almost 0 nm, 290 nm, 697 nm, 926 nm to 1614 nm, for sample I, II, III, IV, and V, respectively) of the double-stranded DNA of the corresponding products (Supplementary Fig. 10c, d). Next, the same amounts by weight of the corresponding products (Fig. 3a) were transfected into mammalian cells using lipofection to evaluate EGFP gene expression, which showed a higher transfection efficiency for Css-dsDNA samples (II–V) when compared to Css DNA (I) (Fig. 3b). Besides, with increasing length of the complementary strand of Css DNA, we observed a trend of significantly increased mean fluorescence intensity (up to twofold) for Css-dsDNA samples (II–V) relative to Css DNA (I) (Fig. 3c). The overall transfection efficiency and expression efficiency of Css-dsDNA samples (II–V) thus depend on the length of a partially long double-stranded DNA hybridization. Css-dsDNA generated by the enzyme

digestion method retained relatively long continuous complementary fragments bound to it, but this did not allow us to explore a potential dependence on the sequence or binding position of the complementary strands.

We therefore next studied the effect of short complementary DNA strands hybridized to different regions on the Css EGFP(+) on its expression efficiency. As shown in Fig. 3d, to this end the whole Css DNA was divided into 84 fragments, corresponding to 84 complementary, 24-nt-long single-stranded DNAs (denoted by numbers 1, 2, 3…84, respectively, sequences are given in Supplementary Table 9). Individual complementary strands were separately added to the Css DNA, and the corresponding hybridization products were transfected into cultured MDCK cells to record the resulting mean fluorescence intensities (Fig. 3e and Supplementary Figs. 11 and 12) and transfection efficiencies (Supplementary Fig. 13). We observed that the corresponding hybridization products had a similar range of transfection efficiencies relative to Css EGFP(+). Interestingly, however, it is found that 45% of the 84 complementary strands resulted in a slight enhancement of Css DNA expression (up to 122% of the EGFP expression efficiency of Css DNA without any complementary strand), while 43% of the strands had a slight inhibitory effect (displaying 80–100% EGFP expression efficiency). Another 10% of the strands had a moderate inhibitory effect (around 60–80% EGFP expression efficiency), and only 2% of the complementary strands reduced expression efficiency more strongly (58% and 55%, respectively). Overall, our results indicate that, when only using a single short complementary strand binding to one specific sequence domain on the Css DNA, gene expression efficiency is either enhanced or reduced depending on the location of the targeted site, but we did not observe a clear correlation between binding site and regulatory effect.

As a next step, we simultaneously added two 24-nt-long complementary strands binding to two different locations on the Css DNA (1 + 43, 2 + 44,… 42 + 84, see Fig. 3f) and studied the effect of these pairs on gene expression. As shown in Fig. 3g and in Supplementary Fig. 14, only 17% of the tested combinations resulted in a slight enhancement of Css DNA expression compared to the control, while the other combinations showed different degrees of inhibition of gene expression. Of these, 60% of the combinations only had a weak inhibitory effect (80% - 100% EGFP expression efficiency), 16% of the strands had a moderate inhibitory effect (60–80% EGFP expression efficiency), while 7% of the combinations reduced expression efficiency more strongly (57%, 47%, and 46%, respectively). Notably, we also observed a reduced transfection efficiency in the last 7% of the combinations relative to the original Css EGFP(+) (Supplementary Fig. 15). Thus, compared to the use of only single 24-nt-long complementary strands, some of the pairwise combinations of the individual strands led to a slightly stronger inhibition of Css DNA expression.

**Suppression of gene expression from Css DNA using blocking strands**

As explained in Fig. 4a, we next fused two 24-nt-long complementary ssDNAs binding to two separate locations on the Css DNA into a single

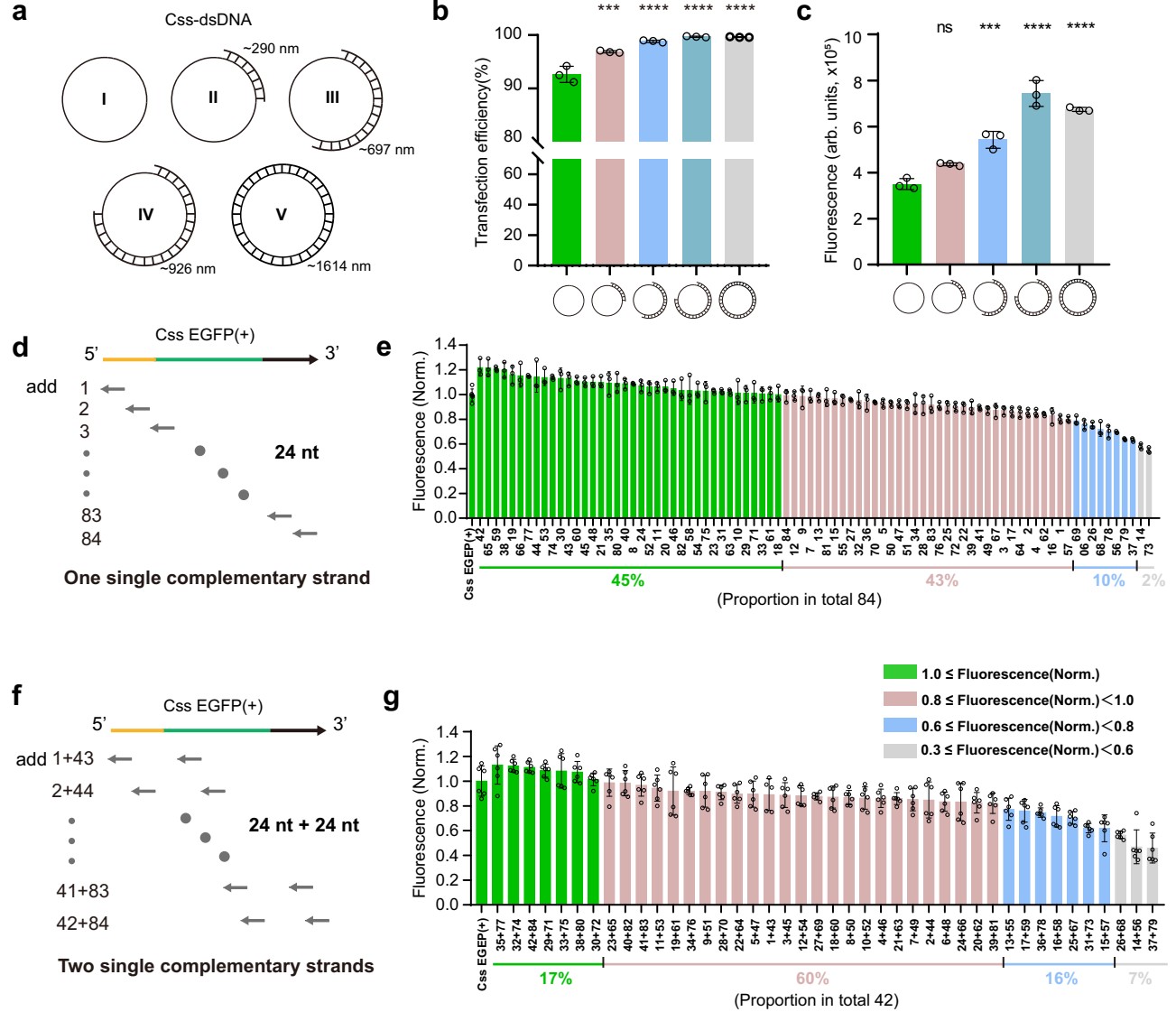

**Fig. 3 | Effect of different complementary strands of Css DNA on gene expression. a** Css DNA with a long complementary strand with different strand length (such as approximately 0 nm, ~290 nm, 697 nm, 926 nm, and 1614 nm for sample I, II, III, IV, and V, respectively) termed hybrid "Css-dsDNA". **b, c** Comparison of transfection efficiency and mean fluorescence intensity of cultured MDCK cells transfected (24 h) with Css-dsDNA (I–V). II ($p = 0.0001$), III ($p = 4.93 \times 10^{-6}$), IV ($p = 1.44 \times 10^{-6}$) and V ($p = 1.58 \times 10^{-6}$) demonstrated higher transfection efficiency, while III ($p = 0.0002$), IV ($p = 2.89 \times 10^{-7}$) and V ($p = 1.77 \times 10^{-6}$) demonstrated higher mean fluorescence intensity (arb. units) compared to I. **d** Schematic showing a total of 84 short strands (24 nt) covering the whole Css DNA were separately added to Css DNA. **e** Single complementary strands with the same length (24 nt) (each 2.5 pmol) were separately added to Css DNA (0.5 pmol), in which the corresponding products were transfected into cultured MDCK cells (24 h) to obtain corresponding mean fluorescence intensities. **f** Schematic showing two single complementary strands with the same length (24 nt) (each 2.5 pmol) located in two different positions of Css DNA were together added to Css DNA (0.5 pmol). **g** The corresponding products (**f**) were transfected into cultured MDCK cells (24 h) to obtain corresponding mean fluorescence intensities. In **e** and **g**, Fluorescence Norm. indicates that all mean fluorescence intensities were normalized to the fluorescence value of the corresponding mammalian cells transfected with untreated Css EGFP(+). Data collected in **b, c, e** and **g** were quantified using flow cytometry and are presented as mean ± standard deviation (s.d.) for $n = 3$ (**b, c, e**) or $n = 6$ (**g**) biologically independent experiments, individual data points are overlaid, source data provided. Statistical analysis in **b** and **c** was performed using one-way ANOVA with Tukey's multiple comparison (***$p \leq 0.001$, ****$p \leq 0.0001$, ns $p > 0.05$).

blocking strand of twice the length. Specifically we created the 48-nt-long combinations 1-43 (strand 1 with strand 43), 2-44,...., 42-84 (Sequences are given in Supplementary Table 10), and explored the effect of these fused blocking strands on the suppression of gene expression from Css DNA. As shown in Fig. 4b and in Supplementary Fig. 16, only 2% of the fused ssDNAs had no inhibitory effect on gene expression at all, while all other tested fused ssDNAs inhibited Css DNA expression to different degrees. Of these, 33% of the fused blocking ssDNAs resulted in only a slightly decreased EGFP expression efficiency (80% -100%), 33% of the fused ssDNAs had a moderate inhibitory effect of ~60–80% EGFP expression efficiency, while 31% of the

fused ssDNAs resulted in a stronger inhibition of Css DNA expression (below 60%, with the lowest at only 37% EGFP expression efficiency). Notably, the latter 31% also displayed an obvious trend towards reduced transfection efficiency (Supplementary Fig. 17).

To further explore the potential sequence dependence of the inhibitory effect of the blocking strands, we designed another set of 48-nt-long blocking strands that cross-linked the Css DNA in a different way. We specifically targeted the CMV promoter, the EGFP coding region and the non-coding region of the Css DNA with blocking strands denoted CB, EB, and NB, respectively. In total, we tested 49 blocking strands for which we systematically varied the position and the

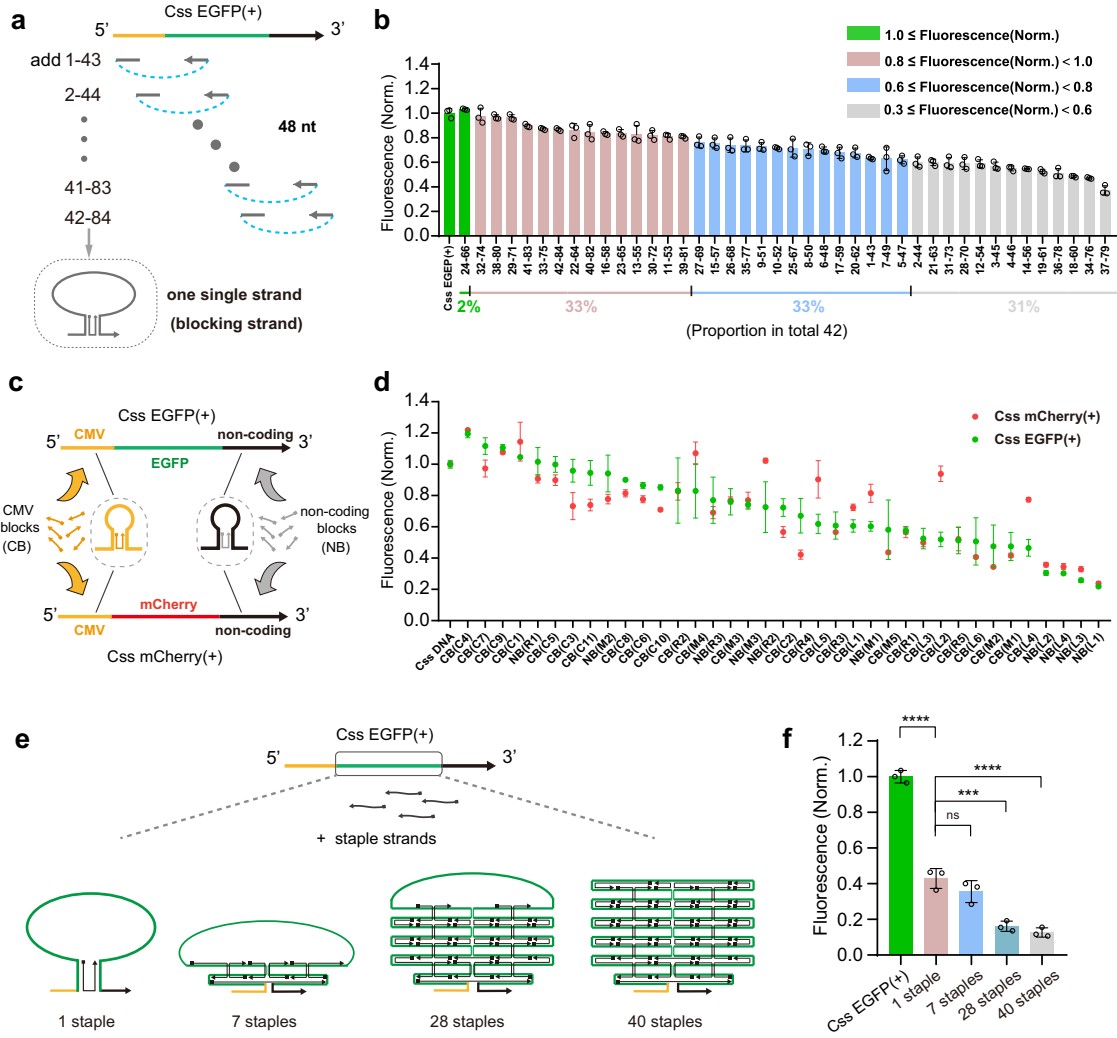

**Fig. 4 | Various blocking strands for suppressing gene expression of Css DNA. a** Schematic showing single fused strands (48 nt) (each 2.5 pmol) which contain two complementary sequences shown in Fig. 3c were separately added to Css DNA (0.5 pmol). **b** The corresponding products in **a** were transfected into cultured MDCK cells (24 h) to obtain corresponding fluorescence intensities. **c** Various blocking strands in CMV promoter (yellow) termed "CMV blocks (CB)" or non-coding region (black) termed "non-conding blocks (NB)" were separately added to Css EGFP(+) and Css mCherry(+), respectively. **d** The corresponding hybridization products in **c** were transfected into cultured MDCK cells for 24 h and to record corresponding mean fluorescence intensities. The dots in green and red are the expression efficiencies of Css EGFP(+) and Css mCherry(+), respectively, affected by the corresponding blocking strand. **e** The construction of different complicated structures on EGFP region of Css EGFP(+) by adding various number staple strands

(1, 7, 28, and 40 staples, respectively). **f** The expression efficiency analysis of cultured MDCK cells transfected with the corresponding products in **e**. Lower fluorescence intensity was observed for 1 staple ($p = 1.67 \times 10^{-7}$) compared to untreated Css EGFP(+). In addition, 28 staples ($p = 1.66 \times 10^{-4}$) and 40 staples ($p = 5.74 \times 10^{-5}$) demonstrated lower fluorescence intensities than 1 staple. Data collected in **b**, **d** and **f** were quantified using flow cytometry and are presented as mean ± standard deviation (s.d.) for $n = 3$ biologically independent experiments, individual data points are overlaid for **b** and **f**, source data provided. In **b**, **d** and **f**, Fluorescence Norm. indicates that all fluorescence intensities were normalized to the fluorescence value of the corresponding mammalian cell transfected with untreated Css EGFP(+). Statistical analysis in **f** was performed using one-way ANOVA with Tukey's multiple comparison (***$p \leq 0.001$, ****$p \leq 0.0001$, ns $p > 0.05$).

distance between the cross-linking positions within the respective domains (Fig. 4c, Supplementary Fig. 18, and sequences are given in Supplementary Table 10). For instance, CB(L1) is a 48 nt blocking strand that binds to 24 nt at the leftmost end of the CMV promoter and cross-links it with another 24-nt-long domain in the promoter, creating a loop of 10 nt length in the Css DNA. In the same way, CB(L2),... CB(L6) bind to the same leftmost domain, but connect them to sequences with increasing distance (and thus create larger loops up to 460 nt length). Correspondingly CB(M1)... CB(M5) bind to the middle portion of the promoter, and CB(R1)... CB(R5) bind to the rightmost part. Following the same scheme, crosslinker strands were also designed for the other regions, i.e., EB(L1)... EB(L4), EB(M1)... EB(M4), EB(R1)... EB(R3), NB(L1)... NB(L4), NB(M1)... NB(M3), NB(R1)... NB(R3). The

corresponding hybridization products were transfected into cultured MDCK cells for 24 h, and we found that 35% of the tested blocking strands had a significant effect on Css DNA expression (below 60%, with the lowest at only 22%, Fig. 4d and Supplementary Fig. 19).

To further explore the generality of the blocking effect, we added the CB and NB blocking strands also to Css mCherry(+) (Supplementary Fig. 20), which has the same CMV region and non-coding region as Css EGFP(+) (Fig. 4c). For most of the blocking strands tested for Css EGFP(+) and Css mCherry(+), we observed a similar trend for the inhibition of gene expression efficiency (Fig. 4d) and transfection efficiency (Supplementary Fig. 21), suggesting that the inhibitory action of blocking strands targeting the same sequence regions is similar. Although the sequences of Css mCherry(+) and Css EGFP(+)

only differ in the gene coding regions (~720 nt) (which are not targeted by the CB and NB strands), the long Css DNA strands themselves (both containing more than 2000 nt in total) have rather complex secondary structures (Supplementary Figs. 22 and 23), which are likely to interfere with the binding of the blocking strands, resulting in different effects on Css DNA expression. We therefore speculate that mode of action of the blocking strands is quite general and consistent, but their inhibitory efficiency is affected by the formation of secondary structures within the long Css DNA.

In order to rule out that the inhibitory effect of the blocking strands is not based on their direct interaction with mRNA, we also performed blocking experiments with the conventional double-stranded plasmid DNA (pl EGFP). We selected three blocking strands (EB(M4), NB(L1) and NB(L3)) and added each of them separately to pl EGFP. The corresponding mixtures were transfected into MDCK cells, the corresponding mean fluorescence intensities and transfection efficiencies were recorded. The three selected blocking strands have a significant inhibitory effect on Css EGFP(+) expression (Supplementary Fig. 24a, b), but do not inhibit expression from pl EGPF (Supplementary Fig. 24c, d), suggesting that the blocking strands do act via direct interaction with mRNA. We were also concerned that unmodified DNA single strands would be quickly degraded in vivo, we also explored two chemical modification approaches for a 48 nt blocking strand (EB(M4)) using phosphorothioate or methylation to form Css EGFP(+) – EB(M4) (phosphorothioate) and Css EGFP(+) – EB(M4)(methylation), respectively. Interestingly, the modification did not improve the inhibitory effect of the blocking strand (Supplementary Fig. 25).

On the whole, fused ssDNAs (48 nt) that crosslink separate sequence domains on the Css DNA typically resulted in a more significant inhibition of Css DNA expression than only combinations of two individual 24-nt-long complementary strands. The 48-nt-long DNA blocking strands can bind to the Css DNA in two distinct DNA crossover geometries. We speculate that such crossover structures might inhibit the DNA replication machinery - required for the conversion of Css DNA into double-stranded form - to different degrees, resulting in the observed differences in gene expression levels.

Inspired by methodologies previously developed in the context of DNA nanotechnology, based on a blocking strand "EB(M4)" (48 nt), we next designed "staple"-like blocking strands with varying lengths (Sequences are given in Supplementary Table 11), consisting of 24 nt, 28 nt, 36 nt, 40 nt, 44 nt, 48 nt, 80 nt, 100 nt, respectively, and evaluated their inhibitory effect on gene expression from Css DNA. The blocking staple strand can form a closed loop by pairing with two separate sequence regions near the 3′ and 5′ ends of the EGFP coding sequence (Css EGFP(+)_1/2/3/4/5/6/7/8). As shown in Supplementary Fig. 26, with increasing length of the blocking strands, the suppression of gene expression becomes significantly more pronounced, with a maximum inhibition of EGFP expression observed for a single 48 nt blocking strand (with only ~32% EGFP expression efficiency compared to Css DNA without any blocking strand), as shown in Supplementary Fig. 27. For the longer blocking strands (80 nt and 100 nt), suppression slightly decreased, which presumably is caused by the formation of secondary structures within the long ssDNA blocking strands themselves (Supplementary Fig. 28).

Rather than generating linear Css DNA complexes with complementary DNA single strands, we next applied methods inspired by DNA origami assembly[23,24,37,38] to construct molecular nanostructures that were folded and compressed in EGFP coding sequence regions, which was expected to limit their accessibility by DNA binding proteins. As shown in Fig. 4e and Supplementary Fig. 29a, for the EGFP coding region, we added a different number of staple strands 1 staple, 7 staples, 28 staples, and 40 staples (sequences are given in Supplementary Table 12) to assemble tightly compressed DNA structures. These four structures were then separately transfected into cultured MDCK cells for culturing of 24 h. With an increasing number of staple

strands in EGFP coding region, respectively, EGFP expression efficiency relative to Css EGFP(+) significantly decreased from 43% (for 1 staple) to 13% (for 40 staples), respectively (Fig. 4f and Supplementary Fig. 29b), and we also observed an obvious trend of reduced transfection efficiency in all cases relative to the original Css EGFP(+) (Supplementary Fig. 30).

**Construction of a gene expression regulator based on Css DNA**

We next applied concepts developed previously in dynamic DNA nanotechnology to render gene expression from Css DNA templates switchable by the addition of appropriate inhibitor and activator strands. We selected 48 nt blocking sequences that had been found to significantly inhibit gene expression from Css DNA (Supplementary Fig. 26) for further experiments. As shown in Fig. 5a, we first introduced 58-nt-long Toe-CB(M1) or Toe-EB(M4) strands to Css EGFP(+) in vitro, each of which could hybridize to 48 nt in the CMV promoter and the EGFP coding region, respectively, and was equipped with an additional 10-nt-long unpaired toehold sequence. Compared to the single blocking strands utilized above, the addition of 10-nt-long toehold sequences were found to slightly affect the inhibition of Css EGFP(+) expression (Fig. 5b). We next explored the potential regulatory effect of complementary trigger strands (CB(M1)T for Toe-CB(M1) and EB(M4)T for Toe-EB(M4)), which were designed to release the corresponding blocking strands from the Css EGFP(+) via toehold-mediated strand displacement (TMSD)[39]. To this end, we complexed the trigger strands with transfection agent and verified that - outside of the cell – they did not interact with lipoplexes containing the blocked Css EGFP(+) DNA (Supplementary Fig. 31). The trigger strands were then co-transfected together with the blocked Css EGFP(+) into cultured MDCK cells.

We observed a similar range of transfection efficiencies for blocked-Css EGFP(+) relative to the corresponding trigger groups (Supplementary Fig. 32), but as shown in Fig. 5b, compared to the blocked Css EGFP(+), a significant recovery (~50−85% EGFP expression efficiency relative to Css EGFP(+) expression) in mean fluorescence intensity was observed in the presence of the CB(M1)T strand for the Toe-CB(M1)-blocked Css EGFP(+) or EB(M4)T strand for the Toe-EB(M4)-blocked Css EGFP(+), indicating that Css DNA-based gene expression was successfully activated by the trigger-induced strand displacement reaction. Further, a 2-input AND gate was successfully constructed (Fig. 5c), in which both the Toe-CB(M1) strand and the Toe-EB(M4) strand were added to Css EGFP(+) together, and the doubly blocked Css EGFP(+) and the corresponding two input triggers (CB(M1)T and EB(M4)T) were co-transfected into MDCK cells. As shown in Fig. 5d, compared to the doubly blocked Css EGFP(+) (with ~33% EGFP expression efficiency relative to Css EGFP(+) expression), a significant recovery in fluorescence intensity (~79% EGFP expression efficiency) was observed in the presence of both CB(M1)T and EB(M4)T strands, and a trend towards increased transfection efficiency was also observed in the presence of both CB(M1)T and EB(M4)T strands (Supplementary Fig. 33), indicating that 2-input AND gate-based Css EGFP(+) expression was successfully activated. In addition, as shown in Fig. 5e and Supplementary Fig. 34, compared to Css mCherry(+) doubly blocked via the Toe-CB(M2) strand and Toe-NB(L2) strands (with ~35% mCherry expression efficiency relative to Css mCherry (+) expression), a significant recovery in mean fluorescence intensity (~67% mCherry expression efficiency) and in transfection efficiency was observed in the presence of both CB(M2)T and NB(L2)T strands (two input triggers), indicating that we also successfully constructed a 2-input AND gate based on Css mCherry(+) expression by the trigger-induced strand displacement reaction.

We finally investigated multi-input AND logic by testing circuits with increasing numbers of blocking strands and inputs. As an example, Fig. 6a presents a schematic of the construction of a 5-input AND circuit. In order to explore the specificity of Css DNA-based gene

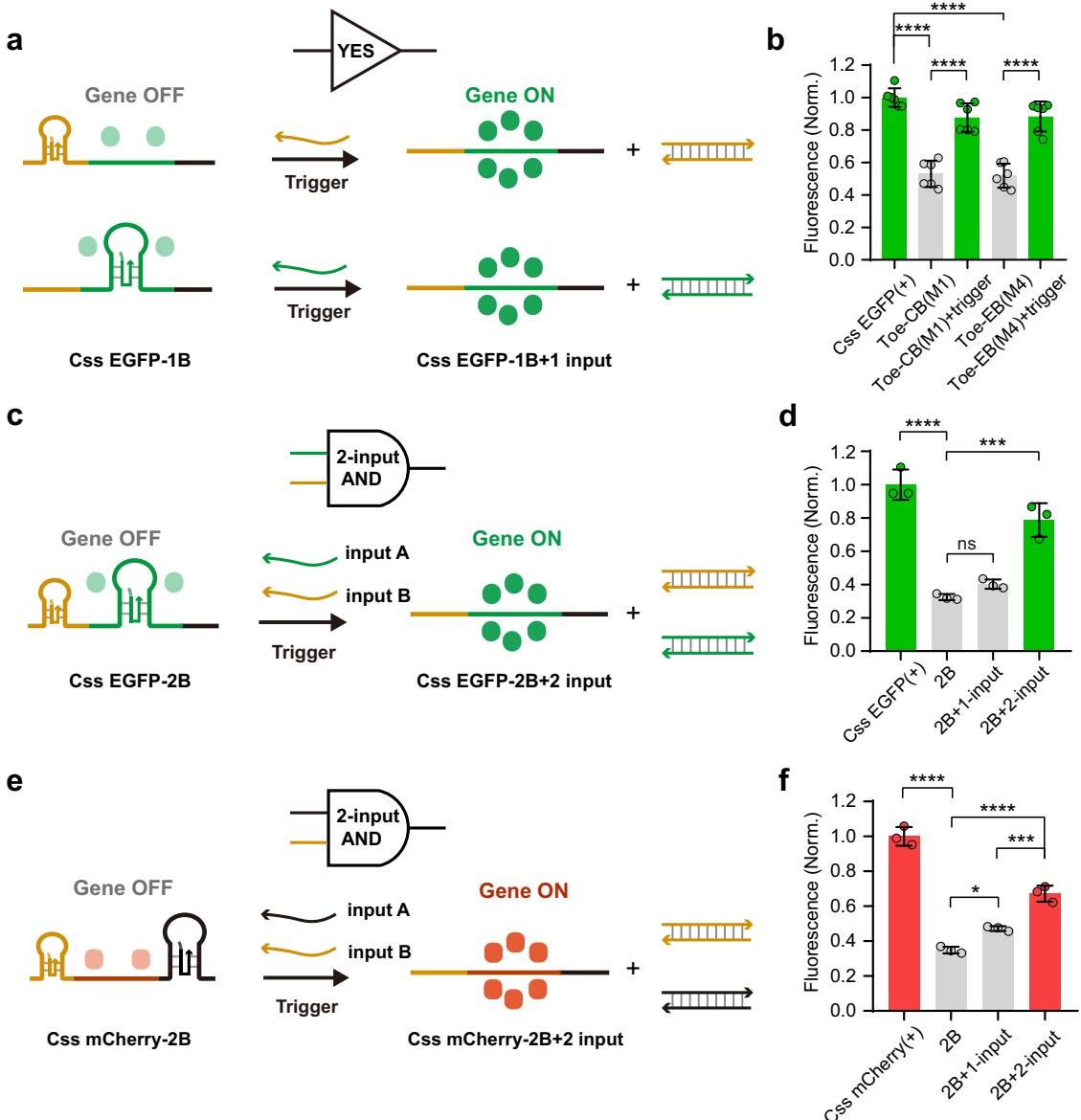

**Fig. 5 | Toehold switch regulator design (2-input AND gate) and characterization in cultured MDCK cells.** The design (**a**) and fluorescence intensities (**b**) of Css EGFP(+) regulator (1-input YES) empowered by adding a single block strand and a toehold-switch trigger strand to CMV or EGFP region via TMSD reaction. Lower fluorescence intensities were observed for Toe-CB(M1) ($p = 2.0 \times 10^{-9}$) and Toe-EB(M4) ($p = 1.2 \times 10^{-9}$) compared to untreated Css EGFP(+). In addition, Toe-CB(M1)+trigger ($p = 6.29 \times 10^{-7}$) and Toe-EB(M1)+trigger ($p = 2.57 \times 10^{-7}$) demonstrated higher fluorescence intensities than Toe-CB(M1) and Toe-EB(M1), respectively. The design (**c**) and fluorescence intensities (**d**) of Css EGFP(+) regulator (2-input AND gate) empowered by adding two single blocking strands and two toehold-switch trigger strands to CMV and EGFP region via TMSD reaction. Lower fluorescence intensity was observed for doubly block (2B) ($p = 1.11 \times 10^{-5}$) compared to untreated Css EGFP(+). In addition, 2B + 2-input ($p = 1.82 \times 10^{-4}$) demonstrated higher fluorescence intensity than 2B. The design (**e**) and fluorescence intensities

(**f**) of Css mCherry(+) regulator (2-input AND gate) empowered by adding two single blocking strands and two toehold-switch trigger strands to CMV and non-coding region via TMSD reaction. Lower fluorescence intensity was observed for doubly block (2B) ($p = 8.45 \times 10^{-8}$) compared to untreated Css mCherry(+). 2B + 1-input ($p = 0.0155$) demonstrated higher fluorescence intensity than 2B, while 2B + 2-input ($p = 2.48 \times 10^{-5}$, $p = 0.0008$) demonstrated higher fluorescence intensities compared to 2B and 2B + 1-input, respectively. Data collected in **b**, **d** and **f** were quantified using flow cytometry and are presented as mean ± standard deviation (s.d.) for $n = 3$ biologically independent experiments, individual data points are overlaid, source data provided. In **b**, **d** and **f**, Fluorescence Norm. indicates that all fluorescence intensities were normalized to the fluorescence value of the corresponding mammalian cell transfected with the corresponding untreated Css DNA. Statistical analysis in **b**, **d** and **f** was performed using one-way ANOVA with Tukey's multiple comparison (*$p \leq 0.05$, ***$p \leq 0.001$, ****$p \leq 0.0001$, ns $p > 0.05$).

expression regulation via the strand displacement reaction, we first conducted an orthogonality test with five blocking strands (Block A, B, C, D, E) and five input strands (Input A, B, C, D, E) (Sequences are given in Supplementary Table 11). In addition, to measure crosstalk between blocking strands and input strands more strictly, blocking strands based on block-input E, but with different toehold sequences (block-input E1 and block-input E2) were also tested for orthogonality. Figure 6b indicates the relatively low levels of crosstalk for 7 × 7 pairwise

combinations of blocking and input strands in cultured MDCK cells. Representative fluorescence images and flow cytometry GFP fluorescence analysis of cultured MDCK cells transfected with unblocked and quintuple blocked Css EGFP(+), in the absence and presence of input strands, respectively, are shown in Fig. 6c, d. As shown in Fig. 6e and Supplementary Fig. 35, we observed a similar trend of enhanced expression efficiency and increased transfection efficiency, and found that with increasing number of blocking strands (5B), the inhibitory

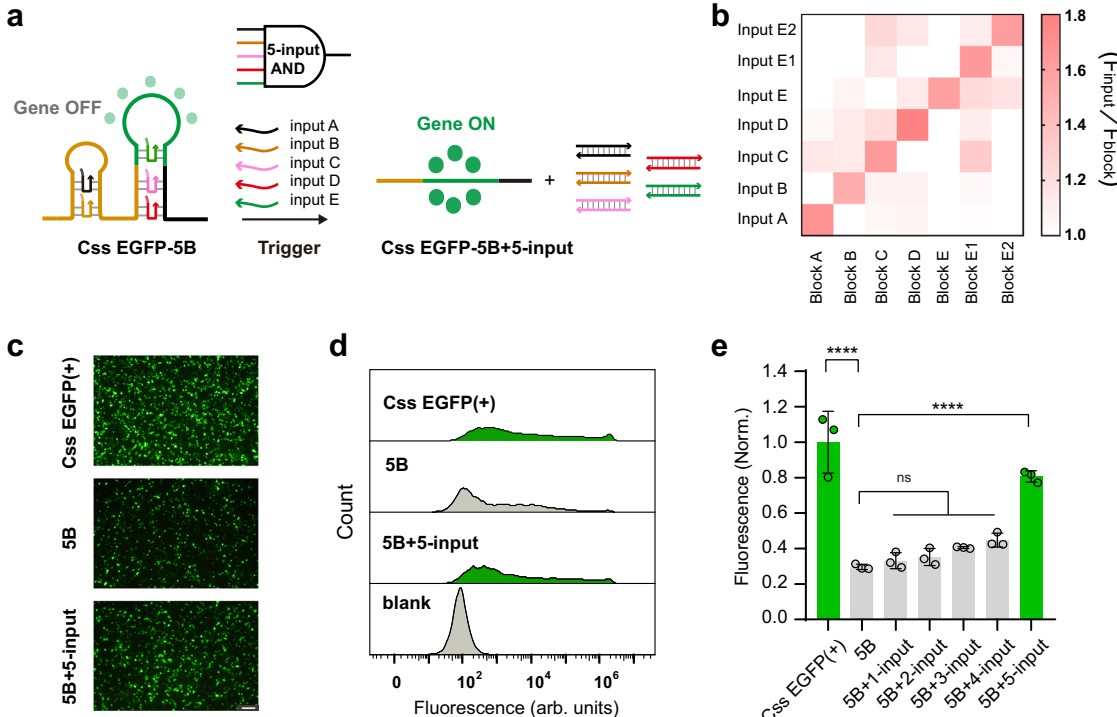

**Fig. 6 | Toehold switch regulator design (5-inputs AND gate) and characterization in cultured MDCK cells. a** The scheme illustration of the circular single-stranded gene expression regulator empowered by adding five single blocking strands and five toehold-switch trigger strands via TMSD reaction (5-input AND gate). **b** Assessment of seven toehold switches orthogonality consisting of block-input A, B, C, D, E, E1, and E2, in which block-input E1 and E2 only change toehold sequence on the basis of block-input E. $F_{input}/F_{block}$ represents EGFP fluorescence ratios of Css EGFP(+)-block with different input signal and Css EGFP(+)-block after 24 h of transfection in cultured MDCK cell. **c** The representative fluorescence images of cultured MDCK cells transfected with single Css EGFP(+), the blocked Css EGFP(+) with 5 blocking strands (5B) and the blocked Css EGFP(+)-5-input trigger strands (5B + 5-input), respectively. The images are representative of one of $n = 3$ biologically independent experiments; similar results were observed each time. Scale bar, 100 μm. **d** The representative flow cytometry GFP fluorescence analysis for five toehold switchable regulator compared to MDCK cell autofluorescence (blank) and a Css EGFP(+) positive control. Autofluorescence level was measured from MDCK cells not treated with Css EGFP(+). The flow analysis is representative of one of $n = 3$ biologically independent experiments; similar results were observed each time. **e** Fluorescence intensity of the circular single-stranded gene expression regulator (5-input AND gate). Fluorescence Norm. indicates that all fluorescence intensities were normalized to the fluorescence value of the corresponding mammalian cell transfected with the corresponding untreated Css EGFP(+). Lower fluorescence intensity was observed for 5B ($p = 1.99 \times 10^{-7}$) compared to untreated Css EGFP( + ). 5B + 5-input ($p = 1.03 \times 10^{-5}$) demonstrated higher fluorescence intensity than 5B. Data collected in **b** and **e** were quantified using flow cytometry and are presented as mean ± standard deviation (s.d.) for $n = 3$ biologically independent experiments, individual data points are overlaid in **e**, source data provided. Statistical analysis in **e** was performed using one-way ANOVA with Tukey's multiple comparison (****$p \le 0.0001$, ns $p > 0.05$).

effect on Css DNA expression is enhanced (~30% EGFP expression efficiency). Conversely, when increasing numbers of input strands are added to quintuply blocked Css EGFP(+) (5B), EGFP expression will significantly increase (up to 81% for the 5B + 5-input case) only in the presence of 5-input (Input A, B, C, D, E, together). In addition, we explored the potential for a long-term dynamic regulation of gene expression, as shown in Supplementary Fig. 36. The lip2000-loaded blocked Css EGFP(+) construct was first transfected into MDCK cells, and the transfection-medium was then removed and replaced with the corresponding lip2000-loaded trigger strands after 4 h of transfection. However, no apparent changes in the EGFP expression levels were observed, possibly due to failure strand displacement in the crowded environment inside the cell. This suggests that this regulatory strategy based on Css DNA system need more efforts to realize the long-term dynamic regulation within the complex cellular environment.

## Discussion

Our work establishes an approach towards synthetic gene regulation based on circular single-stranded DNA (Css DNA). Css DNA can be used as an efficient gene expression vector, whose activity can be controlled by hybridization with partially complementary DNA strands. To demonstrate this capability, we prepared sense and antisense Css DNA (Css EGFP(+) and Css EGFP(−)) in vivo from a recombinant (double-stranded) plasmid hosting the targeted gene fragment. Surprisingly, we found that mammalian cells express the target proteins starting either from sense or antisense Css DNA, suggesting their conversion into double-stranded form in vivo. We then harnessed partially complementary DNA "block" strands to form hairpin structures in the Css DNA strands to hinder the replication of single-stranded DNA and thus inhibit gene expression. Our screening results indicate that not all blocking strands have a similarly strong inhibitory effect. We speculate that blocking strands at different positions affect binding and processivity of enzymes involved in the replication process differently, and thus also result in different expression levels. In addition, we demonstrated that the functionality of blocking strands for suppressing gene expression of Css DNA has a certain generality (Fig. 4d), but it seems to be influenced by the formation of strong secondary structures within the long Css DNA itself. Importantly, addition and removal of blocking strands can be used for the regulation of gene expression from Css DNAs in mammalian cells. Although the long-term dynamic regulation from our Css DNA system was not currently realized, a 5-input AND gate from a Css DNA was constructed, in which gene expression was suppressed by five blocking strands, and recovered by the addition of trigger strands via co-transfection, which remove the blocking strands by a toehold-mediated strand displacement process *in cellulo*.

In structural DNA nanotechnology, circular single-stranded DNAs (such as M13 genomic DNA and other user-customized single-stranded DNAs) have been mostly used as scaffold strands for the construction of DNA origami nanostructures[23,24,37,38] with various applications such as information processing[40–43], spatial organization of proteins[44], nanomedicine[45–48], and genomic integration of nanostructured genes[49]. Recent work[25,26] has reported that gene-encoding DNA origami objects transfected into mammalians cell lines via electroporation would denature in the cell, resulting in expression of the gene encoded the single-stranded scaffold. In contrast, in the present work, we used lipofection as nucleic acid delivery method to transfect Css DNA into mammalian cells. Through systematic screening experiments, we found that already a single strand hybridized to specific sites on the Css DNA can significantly reduce gene expression. Notably, with the addition of more staples expression from the Css DNA becomes increasingly more suppressed (Fig. 4f). We speculate that liposome-coated Css DNA complexes – unlike naked DNA structures delivered by electroporation – are not immediately denatured in the cellular milieu, resulting in a different fate of the Css DNA in the cell. We surmise that replication is most likely initiated by an RNA polymerase, which produces short RNA primers along the Css DNA that allow DNA polymerase to start synthesis of the complementary strand[33]. Presumably, the inhibitory effect of the DNA blocking strands results from an inhibition of this process. An alternative (non-exclusive) mechanism might involve direct transcription of the Css DNA by RNA polymerase, which, however, was previously only reported in the context of rolling circle transcription in bacteria[50,51].

Our Css DNA, which contains gene coding sequences of the target protein, can be transfected into a wide range of mammalian cell lines by lipofection and successfully lead to gene expression (Fig. 2a). We replaced the EGFP fluorescent reporter genes with sequences encoding for other proteins such as mCherry, a combination of mCherry - EGFP, FUS, and luciferase (Fig. 2b–e). Excitingly, protein expression of lipid nanoparticles (LNP)-loaded Css Luciferase can be observed in mice after 12 h intramuscular injection, which may provide potential opportunities for further gene therapeutic applications based on gene-encoding Css DNA in vivo. In principle, Css DNA can thus be utilized as an efficient gene expression vector for the delivery of multiple proteins, maybe even whole pathways or gene circuits, for which trigger DNA (or RNA) would act as a master regulator. Future work will explore more dynamic regulation, such as endogenous regulation by cellularly expressed RNAs as triggers. Apart from nucleic acid-based triggers, various other biologically relevant signals (such as ions, pH, enzymes, or other metabolites) as well as extracellular physicochemical stimuli could be integrated into the Css DNAs by utilizing appropriately functionalized blocking strands, providing the possibility to integrate our Css DNA as gene regulators into more complex and functional synthetic gene networks. Css DNA-based switchable vectors that enable context-dependent, environmentally responsive gene expression expand the repertoire of regulatory mechanisms available for molecular machine and synthetic gene circuitry in mammalian cells with a wide range of potential applications in diagnosis and therapy.

## Methods

### Ethical statement
This research complies with all relevant ethical regulations. All mouse experiments in this research were approved by the Animal Care and Use Committee of Institute of Basic Medicine and Cancer (IBMC), Chinese Academy of Science (2022R0004).

### Strains and growth conditions
DH5α (cat. no. DL1001) and XL1-Blue (cat. no. DL1030) used in this work were purchased from Shanghai Weidi Biotechnology Co., Ltd. DH5α *E. coli* strains were grown in LB medium with ampicillin (10 µg/mL) at 37 °C and 200 rpm. XL1-Blue *E. coli* strains were grown in 2xYT medium with ampicillin (10 µg/mL), chloramphenicol (10 µg/mL) and 500 mM MgCl₂ at 30 °C and 200 rpm.

### Plasmid construction
pScaf-7560.1 (cat. no. 111406 in addgene) was digested by two restriction endonucleases, including KpnI (NEB, cat. no. R3142V) and BamHI (NEB, cat. no. R0136V), to obtain 'pScaf' vector, then we used PCR amplification based on forward primers with a 5' KpnI site and reverse primers with a 3' BamHI site to produce the desired DNA fragments, finally to combine the insert DNAs with the 'pScaf' vector via Hieff Clone® Plus One Step Cloning Kit (Yeasen Biotechnology, cat. no. 10911ES20) following the operation instructions. The constructed corresponding plasmid was sequenced by Tsingke Biotechnology Co., Ltd.

### Construction and synthesis of Css DNA by pScaf phagemid
The pScaf phagemid contains a M13 origin sequence and a modified M13 origin sequence, which served for initiation of circular ssDNA synthesis and as terminator for the rolling circle replication, respectively. With the aid of helper-plasmid (pSB4423) co-transfected in XL1-Blue competent cell, the Css DNA phage can be produced in single-stranded form via intracellular rolling circle amplification and then packaged and harvested in the culture supernatant, which can be purified by standard molecular biology operations for DNA purity. We utilized phagemid technology to generate the Css DNA, such as Css EGFP(+), Css EGFP(−), Css mCherry(+), Css mCherry-EGFP(+), and Css Luciferase(+). Compared with other methods for the synthesis of single-stranded DNA, the pScaf phagemid method has the advantage of having very small fixed sequence regions and can be used to construct and synthesize Css DNA with high purity. Quantification of obtained Css DNA was performed on Nanodrop one (Thermo Fisher). We can currently extract 500 µg of Css DNA with high purity from 1 L of bacterial culture medium (OD = 1.4) via phagemid production.

### Construction and characterization of Css DNA by Exonuclease III digestion
The recognition site sequence "5'-GC↓TGAGG-3'" for the nicking enzyme Nb.BbvCI (NEB, cat. no. R0631S) was inserted into the multiple clone sites of pl EGFP by using the traditional molecular cloning method. First, nicking endonuclease Nb.BbvCI site-specifically cleaved the phosphodiester bond of only one strand of dsDNA pl EGFP (15 µg) in a 50 µL reaction mix consisting of pl EGFP (15 µg), 10x rcut smart buffer (5 µL), nicking enzyme Nb.BbvCI (8.5 µL) and H₂O (added to 50 µL), which were incubated at 37 °C for 2 h, 80 °C for 20 min. Subsequent to Exonuclease III (NEB, cat. no. M0206V) treatment, the nicked strand of pl EGFP-Nb.BbvCI was digested in a 50 µL reaction consisting of pl EGFP-Nb.BbvCI (7.5 µg), 10 x Exonuclease III buffer (5 µL), Exonuclease III (1.5 µL), and H₂O (added to 50 µL) at 28 °C for 5 min; 10 min; 20 min; 30 min or 40 min, after which the reaction was terminated at 65 °C for 5 min. Further, the reaction complexes were purified by SanPrep Column PCR Product Purification Kit. The purified products after Exonuclease III treatment were characterized by atomic force microscope.

### Design and assembly of DNA structures based on Css EGFP(+)
DNA structures were designed using cadnano. All the DNA strands (short-staple DNA strands, sequences are given in Supplementary Table 12) were purchased from Sangon Biotech (Shanghai) Co., Ltd. The corresponding staple strands (final concentration: 40 nM) and Css EGFP(+) (final concentration: 20 nM) were mixed in TAE-Mg²⁺ buffer (40 mM Tris, 2 mM EDTA.Na₂.H₂O, 20 mM acetic acid, 10 mM magnesium acetate, pH 7.4). The DNA samples were heated to 85 °C for 5 min, from 85 °C to 12 °C at the rate of 1 °C/5 min and then 4 °C for 2 h. The formation of DNA nanostructures was finally characterized by a 1% agarose gel.

## Agarose gel electrophoresis

The 1.0% agarose gels with 0.005% (v/v) Ethidium Bromide were prepared by using TAE-$Mg^{2+}$ buffer for DNA structures or TAE buffer for plasmid and Css DNA. Then the assembled DNA samples were run for one hour at a constant 60 V in a TAE-$Mg^{2+}$ buffer, both the plasmids and Css DNAs were run for 35 min at a constant 105 V in a TAE buffer. The gel bands were recorded by Amersham ImageQuant 800.

## AFM imaging

For AFM imaging, samples were prepared by deposition of a 2 μL obtained DNA structure sample onto freshly cleaved mica. After 3 mins, 4 μL water was added onto mica and the mica was blown dry with $N_2$, and the sample was imaged. AFM images were taken in AC air topography mode with an ultra-sharp silicon probe with the spring contant of 0.35 N/m on a Cypher VRS system (Oxford instruments) under ambient condition. All AFM images were analyzed with AR analysis software.

## Cell culture

MC-38 and HCCL-M3 cell lines were kindly provided by Pengfei Zhang (Hangzhou Institute of Medicine, Chinese Academy of Sciences, China). HCT 116 and SW-480 cell lines were kindly provided by Yanlin Song (Xiamen University, China). MDCK cell line was kindly provided by Jinglin Wang (State Key Laboratory of Pathogen and Biosecurity, China), authenticated by STR profiling (FuHeng Biology, China). B16 (CL-0029), MCF-7(CL-0149), AC16 (CL-0790), U87-MG (CL-0238), Hela (CL-0101), HELF (L-0281), HEK-293T (CL-0005), Hepa 1-6 (CL-0105), L929 (CL-0137), A549 (CL-0016), 5637 (CL-0002), Ishikawa (CL-0283), LOVO (CL-0144) cell lines were obtained from Procell Life Science & Technology Co., Ltd (Wuhan, China). HIEC-6 (CRL-3266), WRL-68 (CL-48), THLE-3 (CRL-3583), HUVEC (CRL-1730) cell lines were obtained from the American Type Culture Collection (ATCC). MC-38, B16, HIEC-6, SW480, WRL-68, MCF-7, AC16, THLE-3, U87-MG, Hela, HELF, MDCK, HCT116, HEK-293T, Hepa 1-6, HCCL-M3, HUVEC, L929 and A549 were cultured in Dulbecco's modified Eagle's medium (DMEM) with high-glucose (Gibco, cat. no. 11995040), while 5637 and Ishikawa were cultured in Roswell Park Memorial Institute (RPMI) 1640 medium (Gibco, cat. no. 11875093), LOVO was cultured in Ham's F-12K medium (Gibco, cat. no. 21127022). All cell lines were cultured in medium added with 10% fetal bovine serum (Gibco, cat. no. 10270106) and 1% penicillin/streptomycin (Gibco, cat. no. 15140122) at 37 °C containing 5% $CO_2$.

## In vivo bioluminescence imaging

Five BALB/c mice (6-week-old females) were segmented into three groups, one mouse served as the negative control group, and the rest were evenly divided into plasmid (the positive control) and Css DNA groups. Mice in the experimental groups were injected intramuscularly with 40 μg of nucleic acid substrate (luciferase plasmid and Css DNA), which was encapsulated in LNPs suspended in 50 μL DPBS buffer (pH 7.4). And the negative control group was injected PBS buffer (50 μL). Bioluminescence imaging was performed using the In Vivo Imaging System Lumina (IVIS) Lumina III imaging system (PerkinElmer) after 0.5−47 days. All mouse experiments were approved by the Animal Care and Use Committee of Institute of Basic Medicine and Cancer (IBMC), Chinese Academy of Science(2022R0004). All mice were housed in temperatures 20−25 °C, humidity 30−70%, and a 12 h light/12 h dark cycle.

## DNA transfection by lip2000

Cells were seeded in 24-well plates at a density of 70,000 cells/well and incubated at 37 °C in 5% $CO_2$ for 24 h or 36 h, when the cell density reached ~70−80%, Css DNAs or Css DNAs with hybridization strand (0.5 pmol for each well unless otherwise mentioned) were transfected into the cells by a Lipofectamine 2000 (lip2000; Thermo Fisher, cat.

no. 11668019) transfection reagent (2 μL) following its commercial protocols. Each complex for co-transfection was prepared with lip2000 independently. For 1-input YES, Css EGFP(+) (0.5 pmol) and corresponding single-stranded blocking strand (2.5 pmol) were mixed in TE/10 mM $Mg^{2+}$ (total 50 μL). These DNA samples were annealed 85 °C for 5 min, from 85 to 37 °C at the rate of 1 °C/2 min, 37 °C for 1 h, 12 °C for 2 h, then 2 μL of lip2000 was used for this blocked Css DNA (0.5 pmol) and 1 μL of lip2000 was used for the corresponding trigger strand (50 pmol). For 2-input AND gate, Css EGFP(+) or Css mCherry(+) (0.5 pmol) and two single-stranded blocking strands (each 1.25 pmol) were mixed in TE/10 mM $Mg^{2+}$ (total 50 μL). These DNA samples were annealed by above mentioned method, then 2 μL of lip2000 was used for this blocked Css DNA (0.5 pmol) and 1 μL of lip2000 was used for two trigger strands (each 25 pmol). For 5-input AND gate, Css EGFP(+) (0.5 pmol) and five single-stranded blocking strands (each 1.0 pmol) were mixed in TE/10 mM $Mg^{2+}$ (total 50 μL). These DNA samples were annealed by above-mentioned method, then 2 μL of lip2000 was used for this blocked Css DNA (0.5 pmol) and 1 μL of lip2000 was used for five trigger strands (each 20 pmol). All of transfections were carried out at 37 °C for 4 h, and then the transfection-medium was removed and replaced with fresh growth DMEM containing 10% fetal bovine serum (Gibco, cat. no. 10270106) and 1% penicillin/streptomycin (Gibco, cat. no. 15140122). The transfection results were detected using an inverted fluorescence microscope (Olympus CKX53) and flow cytometry (CytoFLEX LX, Beckman Coulter) after 24 h of transfection.

## Flow cytometry assays

MDCK cells transfected with DNA were cultured for 24 h, and then were collected in 200 μL of phosphate-buffered saline (PBS) buffer for flow cytometry assays. A flow cytometer (CytoFLEX LX, Beckman Coulter) was applied for the measurement of fluorescence intensities by calculating at least 10,000 cells for each measurement. The corresponding fluorometry data were first processed using FlowJo V10 software (exemplary gates are given in Supplementary Fig. 37), and then exported to GraphPad Prism 8.0 for data analysis.

## Definition of transfection efficiency and fluorescence (norm.)

Transfection efficiency: the fraction of cells showing fluorescence intensity higher than that of cells without any treatment via detection of flow cytometry. Fluorescence (Norm.): fluorescence value refers to the mean fluorescence intensity per cell of the whole test group (serve as a proxy for gene expression efficiency), all fluorescence intensities were normalized to the value of MDCK cells transfected with untreated Css DNA.

## PCR amplification

At first, Css EGFP(+) and Css EGFP(−) were transfected into cultured MDCK cells, respectively. Then, after culturing for 12 h, the transfected MDCK cells were treated using a Genomic DNA Mini Preparation Kit (Beyotime Biotechnology, cat. no. D0063) to obtain the corresponding genomic DNA consisting of the Css EGFP(+) group genomic DNA and the Css EGFP(−) group genomic DNA. Besides, genomic DNA was also extracted from blank MDCK cells without any treatment (serving as a blank group genomic DNA). Two primers (primer(+), 50 nt and primer(−), 26 nt) in the CMV promoter region were designed to pair with Css EGFP(+) and Css EGFP(−), respectively. We used a single primer to conduct PCR amplification. The total reaction volume for PCR amplification was 50 μL consisting of primer (10 μM, 2 μL), Taq DNA polymerase (1 μL, Sangon Biotech, cat. no. B500010), 10 x PCR buffer (5 μL), $Mg^{2+}$ (25 mM, 3 μL), dNTP (each 10 mM, 1 μL), target template (10−100 ng), and $H_2O$ (added to 50 μL). Css EGFP(+), Css EGFP(−) group genomic DNA and blank group genomic DNA, respectively, were subjected to PCR amplification with primer(+) following the PCR-1 procedure. Css EGFP(−), Css EGFP(+) group genomic DNA and blank group genomic DNA, respectively, were subjected to PCR

amplification with primer(−) following the PCR-2 procedure. The corresponding primer information and two detailed PCR amplification procedures (PCR-1 and PCR-2) are presented in Supplementary Table 7, respectively. All oligonucleotide sequences in this PCR amplification were purchased from Sangon Biotech (Shanghai) Co., Ltd.

## Statistics and reproducibility
Statistical analyses were performed with GraphPad Prism v8.0 (GraphPad Software Inc.). The data is illustrated as the mean ± standard deviation, and the individual data points representing biological replicates are shown. For all tests, differences were considered significant at $*p \leq 0.05$, $**p \leq 0.01$, $***p \leq 0.001$, $****p \leq 0.0001$. All samples presented in agarose gels are representative of $n = 3$ independent agarose gel electrophoresis repeats. All fluorescence microscopy images of the expression of plasmids or Css DNA are representative of samples imaged on $n = 3$ biologically independent repeats. The Investigators were not blinded to allocation during experiments and outcome assessment.

## Reporting summary
Further information on research design is available in the Nature Portfolio Reporting Summary linked to this article.

## Data availability
The authors declare that the data supporting the findings of this study are available within the article and its Supplementary information file or Source Data file. Source data for each graph has been provided as Source Data 1 (excel), and uncropped gel images have been provided as Source Data 2 (pdf) in the Source Data. Source data are provided with this paper.

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

## Acknowledgements

The authors are grateful for the financial support from the National Natural Science Foundation of China (Nos. 22161132008, to J.S.), the National Key Research and Development Program of China (Grant No. 2021YFF1200200, to J.S.), and the Starry Night Science Fund of Zhejiang University Shanghai Institute for Advanced Study (SN-ZJU-SIAS-006, to J.S.). F.C.S. acknowledges support through the Deutsche Forschungsgemeinschaft (DFG SI 761/5-1) and the Max Planck School Matter to Life.

## Author contributions

J.S., F.C.S., Y.K., and L.T. conceived and designed the study. L.T., Z.T., and J.C. developed the study. Y.L., J.W., and Y.S. characterized DNA samples. L.T. and Z.T. performed the cell experiments. L.T., Y.Z., and J.W. performed the mice experiments. L.T., Z.T., and J.C. analyzed the data and wrote the manuscript. P.Z., Y.K., F.C.S., and J.S. provided project supervision. All the authors discussed the results and approved the final version of the manuscript.

## Competing interests

The authors including J.S., J.C., and L.T. filed a patent for the method and system of nucleic acid delivery based on gene-encoding Css DNA (Application number: PCT/CN2021/121765). The patent covers Css DNA as a gene delivery vector for gene expression in vitro and in vivo for potential nucleic acid drug. The other authors declare no competing interests.
