## [Peer Review File · Nature Communications]

REVIEWER COMMENTS

Reviewer #1 (Remarks to the Author):

The authors presented a novel way to regulate gene expression in mammalian cells, using the principles of synthetic biology and an inspiration from the DNA origami design. As the expression vector, instead of a regular plasmid (circular dsDNA), they used circular single-stranded DNA (C_{ss} DNA). They claim that C_{ss} DNA transforms into its double-stranded counterpart through replication in vivo.

Using C_{ss} DNA vectors allows them to manipulate gene expression in a programmable fashion, using shorter ssDNA strands that interact with the vector. These ssDNA strands resemble staples that hybridize with two different domains of the gene coding region, thus inhibiting the gene expression. By removing these strands via toehold strand displacement reaction, gene expression is again activated. They also created different input AND gates, for the enhanced control. The expression is never completely inhibited so it may be interesting to see how these “leaks” may occur or be improved.

The manuscript is pretty clear written, the idea is nice, and it is evident how it contributes to the current knowledge of gene expression regulation in cells. The research preceding the final design is extensive and well explained. The results arising from this stage can be interesting for the further research into the actual mechanism of gene expression modulation (since at the initial stage they discovered that adding double-stranded regions onto a C_{ss} vector can be either activating or inhibiting, and this effect is not so straightforward). However, I am left with one main question (on which an answer may be obvious, but I do not see it currently): Can the ssDNA strands maybe interact with mRNA thus also possibly influencing expression?

Reviewer #2 (Remarks to the Author):

This manuscript by Tang et al presented an approach of circular ssDNA (C_{ss}DNA)-based molecular and nanoscale (DNA origami) vector for controllable expression of the encoded proteins in mammalian cells. The innovation is high. The data strongly support the conclusions. The manuscript is well presented. While this reviewer is enthusiastic about this manuscript, addressing the following questions would improve the clarification and further understanding of the related studies.

1. C_{ss}DNA-based gene expression system has been reported before (cited in ref 25, 26). Please clarify how the present research has advanced from, or can be differentiated from the prior reported research.
2. what's the production yield in the “large-scale prep of C_{ss}DNA”?

3. please study the biostability of CssDNA.
4. in schematic CssDNA figures, what did the black region represent?
5. CssDNA AFM images showed sizes of 50-100 nm in pretty much all dimensions. Please explain how a relatively small CssDNA (2k nucleotides) can form such big nanoparticles.
6. It was indicated that CssDNA-expressing vector produced CssDNA by rolling circle replication, which would generate concatemeric DNA in which each repeat unit has the same sequence as a CssDNA. Please clarify how such linear concatemeric DNA can be processed to generate CssDNA, and how much yield this process has.
7. in vivo luciferase-coding CssDNA and plasmid were i.m. injected into mice, but their expression patterns appear very different: while plasmids seem to express luciferase at injection site (?), CssDNA expressed it in larger region and likely in liver (Fig. 2e). While this difference doesn't impact the conclusion that CssDNA expressed protein in mice, it'll be helpful to clarify the potential reason for such differences.
8. how long can CssDNA expresses proteins in vitro and in vivo? (12 h post i.m. administration was shown in Fig. 2e).
9. the mechanism of CssDNA replication needs further understanding. While there were some plausible mechanisms discussed in the manuscript, further experimental evidence is needed and practical to deepen the mechanistic understanding. For example
 - a. what's the impact of replicase inhibition or silencing on the dsDNA formation and protein expression from CssDNA?
 - b. are primer DNA or RNA (as in the case of some bacteria) needed for CssDNA replication?
 - c. are there preferable or specific replication initiating sites and stop sites on CssDNA? Are the replicated cDNA circular or linear? Are the replicated cDNA monomer or multimer (if underwent Rolling circle replication)?
 - d. In case the CssDNA replication product is linear, can the replication stop prior to reaching a full circle of CssDNA? Combined with the question of replication stop site possible, what are the size profiles of the protein products, i.e., is the protein product of homogeneous size or does they have a series of different sizes resulting from nick sites within the coding region?
 - e. where is CssDNA replicated? cytosol, nucleus, mitochondria?
10. The lock efficiency of short cDNA to CssDNA is limited. Is the limited biostability of such short cDNA one of the reason? Can biostabilized cDNA help?

Reviewer #3 (Remarks to the Author):

Linlin Tang and co-workers report on a new vector for gene expression using circular single-stranded DNA delivered via lipofectamine transfection. The manuscript is interesting but in my opinion it falls a bit short of conveying the main reason for what the advantage of utilizing ssDNA would be compared to previous forms of gene expression vectors. Overall the manuscript has a lot of data that appears to be hard to interpret into clear conclusions. On the other hand, looking at some work that has very recently been published in Nature Communications, and comparing the results, I think this manuscript constitutes an important complement to that other work, and it is almost concurrent in time, and so this work could also be published in Nature Communications after revision.

Main points:

1. The authors claim that they introduce a regulatory switch mechanic, relies heavily on the results in figs. 5 and 6. However, the strands for regulation are co-transfected with the constructs. In my opinion, the authors should demonstrate more dynamic regulation using either transfection with regulating strands after some time, or via expressed RNA (although the latter could be considered out of scope).
2. The in-vivo transfection is a one-off experiment without biological repeats. I don't think it is necessary to add more repeats but it should in my opinion be moved to the supplement. And importantly, it should come with proper explanation of the entire procedure, see below.
3. From fig. 3c and onwards, the authors switch from reporting transfection efficiency to reporting fluorescence intensity. I think the authors should continue to report transfection efficiency (fraction of fluorescent cells) for all the samples. The value of the mean fluorescence per cell has a quite broad distribution, as can be seen in fig. 6d, and the value of comparing means can be questioned. The transfection efficiency, thus remain an important complement.

Other comments:

Line 110: Why was pl DNA cleaved?

Line 119: Shouldn't this be referencing fig. 1e? Or is this another experiment? If so, I think it should be added to the comparison in 1e.

Line 124. Are there single-stranded DNA viruses? I.e. not bacteriophages, but mammalian single-stranded DNA viruses? If yes, those should be cited. If no, rephrase. (to my knowledge there aren't any, this work makes one assume there should be ssDNA viruses actually)

Line 166 and fig. 2. Some cell lines show significantly higher transformation efficiency for the plasmid form. Authors should comment on this as they have chosen the cell line with the highest efficiency which has about 20% higher efficiency than the second highest and is not a human cell line (MDCK is a canine cell line).

Line 187. The Css-Luciferase shows markedly different biodistribution than the plasmid. The authors should comment on this.

Also:

There are no methods sections for the animal experiments nor the lipid nanoparticle packaging. It is hard for me to judge this experiment without further explanation of what was done, but it appears this was a different delivery mechanism than what is described in the rest of the paper.

Fig. 2e, what does "after i.m." mean?

Line 248 and fig. 3. It seems a bit arbitrary to use a pair of oligos spaced at the half length of the construct. Why not start from single oligos with strong enhancing or inhibitory effect and seeing if the addition of another oligo changes that? Or would there be a synergistic effect between two enhancing oligos or two suppressing oligos? If the authors could provide a rationale for the quite specific lengths chose, that would help the reader. It is also hard to see what conclusions can be drawn from the screening in e. and g.

Fig. 3a. I think the estimated number of bases paired should be shown along the nanometers. A gel of the denatured constructs could even directly show the distribution.

Fig. 3b, should be plotted with a continuous y-axis.

Fig. 4. These results are quite interesting compared to recently published results.

Fig. 4. The DNA-lines should be thicker, colors are hard to see. I don't get the point of the dots in green or red in a hexagonal pattern around the genes to the right of the arrows.

Line 331: The claim that longer blocking strands show decreased inhibition originates from their increased propensity to form secondary structures should be backed by a simulation, at least with e.g. Nupack.

Line 421: "Extremely" seems like an exaggeration as for example block E1/input C is comparable to the correct block - input combinations.

Fig. 5a. The lines are too thin, colors are hard to see and in particular distinguish the different colors for blocking/unblocking.

Fig. 5c. The arrows are a bit misleading. It is as if the images were taken on the same sample, which they are not.

Line 483-485: Can't the oligonucleotides act as primers as well?

Line 494. I think this could be toned down to simply stating that protein expression can be done in-vivo using LNP delivery and CssiDNA. The data is from a single mouse experiment. (Which is also not properly explained, see above)

Response to Reviewer 1

The authors presented a novel way to regulate gene expression in mammalian cells, using the principles of synthetic biology and an inspiration from the DNA origami design. As the expression vector, instead of a regular plasmid (circular dsDNA), they used circular single-stranded DNA (Css DNA). They claim that Css DNA transforms into its double-stranded counterpart through replication in vivo.

Using Css DNA vectors allows them to manipulate gene expression in a programmable fashion, using shorter ssDNA strands that interact with the vector. These ssDNA strands resemble staples that hybridize with two different domains of the gene coding region, thus inhibiting the gene expression. By removing these strands via toehold strand displacement reaction, gene expression is again activated. They also created different input AND gates, for the enhanced control. The expression is never completely inhibited so it may be interesting to see how these “leaks” may occur or be improved.

The manuscript is pretty clear written, the idea is nice, and it is evident how it contributes to the current knowledge of gene expression regulation in cells. The research preceding the final design is extensive and well explained. The results arising from this stage can be interesting for the further research into the actual mechanism of gene expression modulation (since at the initial stage they discovered that adding double-stranded regions onto a Css vector can be either activating or inhibiting, and this effect is not so straightforward). However, I am left with one main question (on which an answer may be obvious, but I do not see it currently): Can the ssDNA strands maybe interact with mRNA thus also possibly influencing expression?

Response: We want to thank the reviewer for the insightful comments and suggestions. In fact, we were initially quite concerned about the signal leakage of the blocking strand system. We thought that these “leaks” might have been caused by the weak biological stability of the blocking strand itself, and therefore replaced it with a phosphorothioate-modified strand. However, we found that the inhibitory effect of the blocking strand was not improved. When we added more strands based on the blocking strand principle (resulting, e.g., in the formation of a folded DNA nanostructure), the inhibitory effect was enhanced. We speculate that the leakage observed even in the presence of blocking strands may be related to the complex secondary structure of the Css DNA itself and its interactions with the complex intracellular environment. Potentially, blocking strands are therefore not entirely bound to the Css DNA, or they are simply displaced by replication-related or other DNA-binding enzymes. On the other hand, the observed leakage, and different blocking efficiencies for various blocking strand designs, also give us the opportunity to speculate about the mechanism of Css DNA expression. Replication of double-stranded DNA proceeds as a semi-conservative process, in which a single strand is used as a template for the synthesis of its reverse complement. To this end, usually a primase (primer enzyme) is required to generate a short RNA primer, which allows DNA polymerase to bind and extend the primer in the 5' to 3' direction. We hypothesize that – in addition to the action of primases - short complementary DNA fragments at specific sites might also act as primers, recruiting DNA polymerases to convert the Css DNA into a double-strand and thus promote gene expression rather than inhibit it. This appears to considerably depend on the position of the DNA fragments, as other short complementary DNA fragments do not seem to act as primers and can thus be identified as blocking strands that interfere with the replication process. We are still in the process of investigating the precise mechanism of Css DNA expression in greater detail.

In response to the reviewer’s main concern, we performed additional experiments to prove that the

blocking strands do not inhibit expression by their potential direct interaction with mRNA. To this end, we selected three blocking strands (EB(M4), NB(L1) and NB(L3)) and transfected each of them separately together with either C_{ss} EGFP(+) or conventional plasmid DNA (pI EGFP) for EGFP expression. The corresponding complexes were transfected into MDCK cells, and the resulting mean fluorescence intensities and transfection efficiencies were recorded. The three selected blocking strands have a significant inhibitory effect on C_{ss} EGFP(+) expression (Figure 1a,b), but do not appear to inhibit EGFP expression from pI EGFP (Figure 1c,d), suggesting that the blocking strands do not inhibit expression by directly interacting with mRNA, but act at the level of the C_{ss} DNA, presumably interfering with the process of converting the C_{ss} DNA from single-stranded to double-stranded form.

Figure 1 (Corresponding to Fig. S24 in the revised Supplementary materials). Comparison of the inhibitory effects of the blocking strands on gene expression. **a** and **b**. Mean fluorescence intensities and transfection efficiencies, respectively, of cultured MDCK cells transfected with the corresponding products (C_{ss} EGFP(+) - blocking strand). **c** and **d**. Mean fluorescence intensities and transfection efficiencies, respectively, of cultured MDCK cells transfected with the corresponding products (pI EGFP - blocking strand). All fluorescence intensities were normalized to the value of the corresponding mammalian cell transfected with C_{ss} EGFP(+) or pI EGFP alone. Error bars represent standard deviations from at least three independent tests. Statistical analysis was performed using one-way ANOVA with Tukey's multiple comparison (*p ≤ 0.05, **p ≤ 0.01, ***p ≤ 0.001, ****p ≤ 0.0001, ns p > 0.05).

Response to Reviewer 2

This manuscript by Tang et al presented an approach of circular ssDNA (CssDNA)-based molecular and nanoscale (DNA origami) vector for controllable expression of the encoded proteins in mammalian cells. The innovation is high. The data strongly support the conclusions. The manuscript is well presented. While this reviewer is enthusiastic about this manuscript, addressing the following questions would improve the clarification and further understanding of the related studies.

1. CssDNA-based gene expression system has been reported before (cited in ref 25, 26). Please clarify how the present research has advanced from, or can be differentiated from the prior reported research.

Response: Thank you for the suggestion. There are mainly the following four points that can be differentiated from previously published articles. 1, The mentioned publications (Nat. Commun. 2023, 14, 1017; J. Am. Chem. Soc. 2023, 145, 4946-4950) present the very interesting use of Css DNA in the form of DNA origami as vectors for protein expression, but how Css DNA actually results in protein expression has not been properly addressed in these papers. Based on our data, a clearer picture can be drawn, and our experiments strongly suggest that Css DNA first needs to be replicated into dsDNA and then follows the central dogma.

2, The papers (Nat. Commun. 2023, 14, 1017; J. Am. Chem. Soc. 2023, 145, 4946-4950) used electroporation to transfect cells with DNA origami and demonstrated that folding the Css DNA into compact structures with a large number of staple strands would not disturb gene expression. In contrast, when using lipofectamine, the phenomenon appears to be markedly different - protein expression is dramatically affected by the presence of staple strands, even when using only a single staple. This suggests that the electroporation method used in the cited studies might destabilize the origami structures, so that these structures are actually delivered to the cell in an unfolded state (and expression is thus not inhibited). Origami destabilization by electroporation in conventional folding buffer has been reported previously by one of us (Chopra et al., Nano Letters 16, 6683–6690 (2016)).

3, We have carried out further experiments to test the universality of Css DNA as a template for protein expression. The efficient expression of Css DNA with different target genes has been demonstrated in various cell lines, and even in mice. We are confident that our experiments will therefore encourage more researchers to explore further applications of the technique.

4, Our work does not focus on the use of origami-folding of gene encoding scaffolds, but explores the selective blocking, cross-linking and unblocking of single-stranded, circular templates in general, which also gives rise to novel opportunities for the regulation of gene expression. In this context, we have also constructed a 5-input AND gate from Css DNA to demonstrate the application of gene regulatory circuits with Css DNA, which is a completely different aspect not explored by the cited studies.

2. what's the production yield in the "large-scale prep of CssDNA"?

Response: Css DNA can be obtained efficiently via phagemid production. We can currently extract 500 ug of Css DNA with high purity from 1L of bacterial culture medium (OD=1.4) via phagemid production.

3. please study the biostability of CssDNA.

Response: We agree with the reviewer's suggestion. As the reviewer suggested, we explored the biostability of C_{ss} DNA and EGFP-mRNA (a linear single-stranded mRNA for EGFP expression) in DMEM medium supplemented with 10% FBS. C_{ss} DNA and EGFP-mRNA were separately added to 20 μ L of DMEM with 10% FBS (both at a weight of 400 ng), and the corresponding samples were incubated at 37°C for a given time (0 h, 0.5 h, 1 h, 2 h, 4 h, 8 h, 12 h, 24 h, 36 h, 48 h and 72h, respectively). The analysis of all samples was conducted with a 1.0 % agarose gel, which was run for 1 h at a constant voltage of 80 V in TAE buffer. As shown in Figure 2, C_{ss} EGFP(+) was stable for up to 2 h, and then gradually degraded until it was completely degraded after 36 h. By contrast, EGFP-mRNA was completely degraded after 0.5 h, even the sample at 0 h without treatment in DMEM with 10% FBS would undergo a small amount of degradation. The results demonstrate that C_{ss} EGFP(+) is more stable than EGFP-mRNA in DMEM medium with 10% added FBS at 37°C.

Figure 2 (Corresponding to Fig. S3 in the revised Supplementary materials). Stability of C_{ss} EGFP(+) and EGFP-mRNA incubated in DMEM with 10% FBS at 37°C.

4. in schematic C_{ss}DNA figures, what did the black region represent?

Response: The black region represents a fixed-sequence region of C_{ss} DNA that is required for the pScaf phagemid method, which contains a modified host origin of replication (ori) sequence and a pScaf ssDNA synthesis terminator (Syn. Biol. 2018, 3, ysy015). We have now added the corresponding description to the revised main text.

5. C_{ss}DNA AFM images showed sizes of 50-100 nm in pretty much all dimensions. Please explain how a relatively small C_{ss}DNA (2k nucleotides) can form such big nanoparticles.

Response: In principle, the distance between base-pairs in double-stranded DNA is 0.34 nm, while the distance between nucleotides in ssDNA is a little larger (a typical range that is assumed is ~ 0.5 nm – 0.7 nm). C_{ss} EGFP(+) contains 2k nucleotides, but is partially double-stranded due to secondary structure formation, which is why, in principle, we cannot treat it as simple random coil structure. Nevertheless, assuming that it actually is a random coil made from double-stranded DNA of half the length (1000 bp), we would get a typical end-to-end distance of $R \sim \sqrt{2 \cdot L_p \cdot 1000 \cdot 0.34 \text{ nm}} \sim 180 \text{ nm}$ (with a persistence length of $L_p=50 \text{ nm}$). On the other hand, when we assume that is a random coil of single stranded DNA, we would get $R \sim \sqrt{b \cdot 2000 \cdot 0.6 \text{ nm}} \sim 50 \text{ nm}$ (with a Kuhn length of $b=2 \text{ nm}$). AFM images represent snapshots of various conformations the DNA might take in solution, and thus the observations are completely in line with these expectations. Note that these are not condensed nanoparticles (no LNP or

transfection agent present).

6. It was indicated that C_{ss}DNA-expressing vector produced C_{ss}DNA by rolling circle replication, which would generate concatemeric DNA in which each repeat unit has the same sequence as a C_{ss}DNA. Please clarify how such linear concatemeric DNA can be processed to generate C_{ss}DNA, and how much yield this process has.

Response: In our study, we produced C_{ss} DNA by using the pScaf phagemid method reported by Douglas et al. (Syn. Biol. 2018, 3, ysy015). The rolling circle reaction we referred to in the manuscript is a reaction in bacteriophage for plasmid replication. In summary, there are other phage proteins and mechanisms in bacteriophage that terminate and circularize the DNA product to intended plasmid with a single repeat. This is different from cell-free rolling circle reaction where the product can be concatemeric DNA, as the reviewer pointed out. More details of the pScaf method can be found in Douglas's publication. In our experiments, we can currently extract 500 ug of C_{ss} DNA with high purity from 1L of bacterial culture medium (OD=1.4) via phagemid production.

7. *in vivo* luciferase-coding C_{ss}DNA and plasmid were *i.m.* injected into mice, but their expression patterns appear very different: while plasmids seem to express luciferase at injection site (?), C_{ss}DNA expressed it in larger region and likely in liver (Fig. 2e). While this difference doesn't impact the conclusion that C_{ss}DNA expressed protein in mice, it'll be helpful to clarify the potential reason for such differences.

Response: We would like to thank the reviewer for this observation. As the reviewer pointed out, a higher level of luciferase expression was observed in liver of mice for C_{ss} DNA, compare to plasmid DNA, which expressed luciferase mainly at the injection site. We believe this phenomenon is probably due to the difference in stability of C_{ss} DNA and plasmid DNA in liver. We have added some discussion in the revised main text to address this point: "We surmise that some of the LNPs would enter the liver for both C_{ss} DNA and plasmid DNA. However, since the plasmid Luciferase (5984 bp) has the bacterial backbone sequence (required for the production of plasmid DNA), such as an antibiotic resistance gene, an origin of replication, etc., the plasmid may be more easily degraded and cleared by the liver, leading to comparatively stronger luciferase expression from C_{ss} DNA within 12 h."

8. how long can C_{ss}DNA expresses proteins *in vitro* and *in vivo*? (12 h post *i.m.* administration was shown in Fig. 2e).

Response: For *in vitro* experiments, we tested gene expression of C_{ss} EGFP(+) in MDCK cells. We used lip2000 to deliver C_{ss} EGFP(+) into MDCK cells, then detecting fluorescence after 4 h, 10 h, 16 h, 20 h, 24 h and 36 h of culturing. We observed a trend of increased gene expression from 4 h to 24 h of transfection (Figure 3A). As shown in Figure 3B & C, an appreciable level of EGFP gene expression was achieved in MDCK cells transfected with C_{ss} EGFP(+), which had a high transfection efficiency (~95%). However, EGFP fluorescence intensity at 36 h of culturing is slightly lower than that at 24 h of culturing, which may be due to protein degradation.

Figure 3. Gene expression of Csx EGFP(+) in MDCK cells at different transfection time. A. The representative fluorescence images of cultured MDCK cells transfected with Csx EGFP(+) at different transfection time. Scale bar, 100 μ m. B and C. Transfection efficiencies and Fluorescence intensities of cultured MDCK cells transfected with Csx EGFP(+) at different transfection time (24 h and 36 h, respectively). Error bars represent standard deviations from at least three independent tests.

For *in vivo* experiments, in our another on-going project mainly focused on the *in vivo* expression of Csx DNA, we explored the possibility of long-term expression of Csx DNA in mice. *In vivo* luminescence images of mice were taken from 0.5 to 47 days after intramuscularly injected (i.m.) with Csx Luciferase loaded into lipid nanoparticles (LNP). A physiological saline group and a LNP-plasmid-Luciferase group were used as a negative control and a positive control, respectively, for luciferase expression. As depicted in Figure 4, Csx DNA encoding luciferase results in an appreciable expression in liver of mice, and even had a higher protein expression level 12 h after injection compared to the positive control group (although the plasmid was expressed mainly at the injection site). We surmise that some of the LNPs would enter the liver for both Csx DNA and plasmid DNA. However, since the plasmid Luciferase (5984 bp) has the bacterial backbone sequence (required for the production of plasmid DNA), such as an antibiotic resistance gene, an origin of replication, etc., the plasmid may be more easily degraded and cleared by the liver, leading to comparatively stronger luciferase expression from Csx DNA within 12 h. In addition, as shown in Figure 4, however, it is found that with increasing time from 0.5 to 47 days, the expressed protein in liver would be metabolically cleared, and both gene expression of Csx DNA and plasmids then mainly occurred at the injection site.

Figure 4 (Corresponding to Fig. S9 in the revised Supplementary materials). In vivo luminescence images of mice after intramuscular administration of physiological saline (control), LNP-plasmid and LNP-Css Luciferase, respectively, for 0.5 - 47 days. **a** and **b** two independent tests for in vivo experiments. The LNPs were injected into the right thigh muscle. The Ciss Luciferase was mainly expressed in liver of mice, the plasmid was mainly expressed at the injection site 12 h after injection, but both gene expression of Ciss DNA and plasmids then mainly occurred at the injection site from 2 – 47 days.

9. the mechanism of CissDNA replication needs further understanding. While there were some plausible mechanisms discussed in the manuscript, further experimental evidence is needed and practical to deepen the mechanistic understanding. For example

a. what's the impact of replicase inhibition or silencing on the dsDNA formation and protein expression from CissDNA?

Response: In cell experiments, we used Aphidicolin to inhibit DNA polymerase in the MDCK cells, because aphidicolin is a regular inhibitor of DNA polymerase α and δ . The MDCK cells were treated in DMEM complete medium supplemented with 15 μ M aphidicolin for 24 h. Then, Ciss EGFP(+)(0.5 pmol) was transfected into MDCK cells by lip2000. After 24 h, the cells were imaged with a fluorescence microscope. MDCK cells transfected with Ciss EGFP(+) and not treated with aphidicolin were used as a control. As shown in Figure 5a, we observed a significantly lower EGFP expression in the MDCK cells treated with aphidicolin than in the untreated cells. However, aphidicolin can serve as a chemical arresting agent, which can prevent mitotic cell division by interfering DNA polymerase activity. Recent reports (J. Am. Chem. Soc. 2023, 145, 4946–4950; J. Am. Chem. Soc. 2023, 145, 9343–9353) have demonstrated that this would also incur a decrease in gene expression, as the arrest also inhibits passive nuclear transport during mitosis. Thus, the above results cannot unequivocally determine whether the decrease in Ciss DNA expression is due to the inhibition of DNA polymerase or the cell cycle inhibition.

Therefore, we verified the effect of DNA polymerase inhibition on C_{ss} DNA expression in an in vitro expression system, for which we constructed a dedicated cell-free C_{ss} DNA (denoted by C_{ss} EGFP(+)_cf). As shown Figure 5b, the expression of C_{ss} EGFP(+)_cf was significantly reduced with increasing aphidicolin concentration, proving that the inhibition of DNA polymerase most likely prevented the formation of dsDNA from the C_{ss} DNA, thereby weakening gene expression. (Please note that the experimental results of our in vitro expression system based on C_{ss} DNA are to be published later.)

Figure 5. Effect of inhibition of DNA polymerase on expression of C_{ss} DNA. (a) In MDCK cells. (b) In cell-free expression system.

b. are primer DNA or RNA (as in the case of some bacteria) needed for C_{ss}DNA replication?

Response: This is a very interesting question that it is, however, very difficult to experimentally assess for us at the moment. Based on the biochemistry of DNA replication and DNA polymerization, there should be a primer as all known DNA polymerases start at the 3'OH of a double-stranded primer. The primers needed for the replication of C_{ss} DNA might be either RNA primers or DNA primers.

For instance, during genome replication, double-stranded DNA genomes have to be unwound to serve as templates for semi-conservative replication, starting from RNA primers generated by dedicated RNA polymerases (primases, e.g., the primase subunit of Pol alpha). Thus, one potential replication pathway is the production of RNA primers from the C_{ss} DNA template (potentially in region that contains double-stranded secondary structure), and these RNA primers would then initiate replication of the C_{ss} DNA based on these RNA primers.

Another possibility is that DNA single strands binding to the C_{ss} DNA might act as primers themselves. As shown in Fig. 3e in the revised manuscript, we found that the addition of double-stranded regions to a C_{ss} DNA vector can either activate or inhibit gene expression, depending on the position. We therefore hypothesize that short complementary DNA fragments at specific sites can act as DNA primers, recruiting the appropriate DNA polymerases to convert the C_{ss} DNA into double-stranded form and therefore promote gene expression. Other complementary DNA fragments at certain sites apparently do not act as primers and rather act as blocking strands that can interfere with the replication process. We are still in the process of investigating the expression mechanism of C_{ss} DNA in greater detail.

c. are there preferable or specific replication initiating sites and stop sites on C_{ss}DNA? Are the replicated cDNA circular or linear? Are the replicated cDNA monomer or multimer (if underwent Rolling circle replication)?

Response: While the bacterial genome is replicated from a single origin, eukaryotic genomes are replicated by many origins of replication. Mammalian genomes are replicated by 30,000–50,000 origins, each activated at a characteristic time during the S phase (Nat. Rev. Mol. Cell Biol. 2010, 11, 728–738). The origins are not determined by a consensus sequence, but rather by contextual cues (PLoS Genet. 2019, 15, e1008320), including vicinity to TSSs (transcription start sites), CpG islands, nucleosome-free regions, G4 quadruplexes, and accessible chromatin (Nat. Rev. Mol. Cell Biol. 2015, 16, 360–374; BMC Biol. 2023, 21, 41). Thus, we surmise that the replication process of our C_{ss} DNA in mammalian cells may not have preferred or specific replication initiation and termination sites, which may be mainly based on the contextual cues, different gene-coding C_{ss} DNA may therefore have different replication initiation and termination sites (these might also be related to the secondary structure in the C_{ss}DNA, see also b.).

In addition, the paper (PNAS, 1980,77,4147-4151) has reported that ³²P-labeled circular single-stranded DNA can be converted into circular double-stranded DNA (closed circular supercoils) in the oocyte nucleus by injecting single-stranded DNA templates into *Xenopus* oocyte nuclei. Thus, we surmise that the replicated cDNA is also closed circular in our C_{ss} DNA.

d. In case the C_{ss}DNA replication product is linear, can the replication stop prior to reaching a full circle of C_{ss}DNA? Combined with the question of replication stop site possible, what are the size profiles of the protein products, i.e., is the protein product of homogeneous size or does they have a series of different sizes resulting from nick sites within the coding region?

Response: As in the response to the previous question, we surmise that the replicated DNA also is in closed circular form. In addition, as shown in Fig. 1d, e in the revised manuscript, compared to the plasmids pl EGFP(+) and pl EGFP(-), C_{ss} EGFP(+) and C_{ss} EGFP(-) had a similarly high EGFP expression efficiency, thus we also surmise that correctly expressed and folded EGFP is the main protein product for our C_{ss} EGFP(+) and C_{ss} EGFP(-).

e. where is C_{ss}DNA replicated? cytosol, nucleus, mitochondria?

Response: This is an interesting question. Generally, DNA polymerases are localized in the nucleus or in mitochondria (DNA polymerase gamma), but they are initially produced, of course, in the cytoplasm. So in principle, C_{ss} DNA could be replicated in all three suggested compartments (given a primer, see above). Based on previous reports in the literature and our own experimental findings, we infer, however, that C_{ss} DNA is most likely replicated within the nucleus. Specifically:

1, A publication in PNAS, 1980,77,4147-4151 has reported that single-stranded DNA can be converted into double-stranded DNA in the oocyte nucleus by injecting single-stranded DNA templates into *Xenopus* oocyte nuclei. In contrast, the replication did not occur in the cytoplasm.

2, A recent publication (JACS, 2023,145, 4946-4950) has reported that inclusion of DTS (DNA nuclear targeting sequences) within gene-encoding DNA origami can enhance active nuclear import, resulting in the promotion of its expression. In addition, as shown in Figure 6, we found that gene expression of C_{ss} DNA preferentially occurs during periods of cell division. The above results demonstrate that the amount and speed of C_{ss} DNA entering the nucleus would affect the gene expression of C_{ss} DNA.

Figure 6. Gene expression of Csx EGFP(+) in MCF-7 cell line was continuously recorded under the same field of view.

3. In addition, as shown in above Figure 5b, addition of Csx DNA to a cell-free expression system confirmed that the inhibition of DNA polymerase α via aphidicolin caused a decrease in gene expression from the Csx DNA. DNA polymerase α is mainly found in the nucleus in eukaryotic cell lines, indicating that Csx DNA is most likely replicated within the nucleus.

10. The lock efficiency of short cDNA to CsxDNA is limited. Is the limited biostability of such short cDNA one of the reason? Can biostabilized cDNA help?

Response: We thank the reviewer for this comment. In order to address this point, we modified the single 48 nt blocking strand (EB(M4)) using phosphorothioate and methylation, respectively, to improve its biological stability, and then Csx EGFP(+) – EB(M4)(phosphorothioate) and Csx EGFP(+) – EB(M4)(methylation) were formed. As shown in Figure 7a, b, it is found that the 48 nt long single blocking strands with or without chemical modification present little difference with respect to their inhibitory effect on EGFP expression.

Figure 7 (Corresponding to Fig. S25 in the revised Supplementary materials). Studying the influence of the chemical modification of 48 nt long single blocking strands of Csx EGFP(+) on EGFP expression. **a** and **b**. Mean fluorescence intensities and transfection efficiencies, respectively, of cultured MDCK cells transfected with the corresponding products (Csx EGFP(+) - blocking strand). All fluorescence intensities were normalized to the value of the corresponding mammalian cell transfected with Csx EGFP(+). Error bars represent standard deviations from at least three independent tests. Statistical analysis was performed using one-way ANOVA with Tukey's multiple comparison (* $p \leq 0.05$, ** $p \leq 0.01$, *** $p \leq 0.001$, **** $p \leq 0.0001$, ns $p > 0.05$).

Response to Reviewer 3

Linlin Tang and co-workers report on a new vector for gene expression using circular single-stranded DNA delivered via lipofectamine transfection. The manuscript is interesting but in my opinion it falls a bit short of conveying the main reason for what the advantage of utilizing ssDNA would be compared to previous forms of gene expression vectors. Overall the manuscript has a lot of data that appears to be hard to interpret into clear conclusions. On the other hand, looking at some work that has very recently been published in Nature Communications, and comparing the results, I think this manuscript constitutes an important complement to that other work, and it is almost concurrent in time, and so this work could also be published in Nature Communications after revision.

Main points:

1. The authors claim that they introduce a regulatory switch mechanic, relies heavily on the results in figs. 5 and 6. However, the strands for regulation are co-transfected with the constructs. In my opinion, the authors should demonstrate more dynamic regulation using either transfection with regulating strands after some time, or via expressed RNA (although the latter could be considered out of scope).

Response: We would like to thank the reviewer for this comment. In order to address this point, we used three "Css EGFP(+)_block_trigger" systems (CB(M1)_trigger; CB(M2)_trigger and EB(M4)_trigger, respectively) to explore the potential for a more dynamic regulation of gene expression. The three Css EGFP(+)_block constructs were separately transfected into MDCK cells using lip2000, which was carried out at 37°C for 4 h, after which the transfection-medium was removed and replaced with the corresponding lip2000-coated trigger strands. The transfection results were monitored using flow cytometry after 24 h. As a control, we co-transfected Css EGFP(+)_block and trigger strands simultaneously. As shown in Figure 8a, all transfections had a high transfection efficiency (above 95%). However, the mean fluorescence intensities (Figure 8b) indicate a difference in gene expression efficiency. Css EGFP(+)_block had 35% - 50% EGFP expression efficiency relative to the original Css EGFP(+) expression. When co-transfected with the corresponding trigger strand, Css EGFP(+)_block could obtain ~ 80% EGFP expression efficiency. By contrast, when the corresponding trigger strands are added later, after 4 h-transfection, there was no apparent change in the EGFP expression level. We surmise that the Css DNA_block strands after 4 h-transfection may have two possible fates. A small fraction of the Css DNA_block strands will be successfully converted into dsDNA form and thus result in protein expression, while the rest will be metabolized and degraded in the cellular environment during this period. This will diminish switchability of the system after such a time lag. This suggests that our regulatory Css DNA_block system may in fact not be suitable for long-term dynamic regulation within the complex cellular environment, when several inputs have to be externally added in sequence.

In addition, as the reviewer suggested, we also think using the expressed RNA to conduct the regulation is a good suggestion. However, the endogenous regulation study is considered out of scope, we would carry out this endogenous regulation study in future experiments.

Figure 8. Transfection efficiencies (a) and mean fluorescence intensities (b) of cultured MDCK cells transfected (24 h) with different Ccss DNA-block systems.

2. The in-vivo transfection is a one-off experiment without biological repeats. I don't think it is necessary to add more repeats but it should in my opinion be moved to the supplement. And importantly, it should come with proper explanation of the entire procedure, see below.

Response: We fully agree with the reviewer. The biological repeat of the in vivo transfection (and we now observe the long-term expression of Ccss DNA) is added to the supplementary materials (see our reply to Referee 2 on question #8 for in vivo experiments). As depicted in Figure 4, the Ccss Luciferase was mainly expressed in liver of mice, but the plasmid was mainly expressed at the injection site 12 h after injection. We have added some discussion in the revised main text to address this point. We surmise that some of the LNPs would enter the liver for both Ccss DNA and plasmid DNA. However, since the plasmid Luciferase (5984 bp) has the bacterial backbone sequence (required for the production of plasmid DNA), such as an antibiotic resistance gene, an origin of replication, etc., the plasmid may be more easily degraded and cleared by the liver, leading to comparatively stronger luciferase expression from Ccss DNA within 12 h. In addition, as shown in Figure 4, it is found that with increasing time from 0.5 to 47 days, the expressed protein in liver would be metabolically cleared, and both gene expression of Ccss DNA and plasmids then mainly occurred at the injection site.

In addition, the corresponding experimental details have been added to the methods section in the revised manuscript as follows: "In vivo bioluminescence imaging: Five BALB/c mice (6-week-old females) were segmented into three groups, one mouse served as the negative control group, and the rest were evenly divided into plasmid (positive control) and Ccss DNA groups. Mice in the experimental groups were injected intramuscularly with 40 μ g of nucleic acid substrate (luciferase plasmid and Ccss DNA), which was encapsulated in LNPs suspended in 50 μ L DPBS buffer (pH 7.4). And the negative control group was injected PBS buffer (50 μ L). Bioluminescence imaging was performed using the In Vivo Imaging System Lumina (IVIS) Lumina III imaging system (PerkinElmer) after 0.5 – 47 days. All mouse experiments were approved by the Animal Care and Use Committee of Institute of Basic Medicine and Cancer (IBMC), Chinese Academy of Science(2022R0004)."

Figure 4 (Corresponding to Fig. S9 in the revised Supplementary materials). In vivo luminescence images of mice after intramuscular administration of physiological saline (control), LNP-plasmid and LNP-C55 Luciferase, respectively, for 0.5 - 47 days. **a** and **b** two independent tests for in vivo experiments. The LNPs were injected into the right thigh muscle for. The C55 Luciferase was mainly expressed in liver of mice, the plasmid was mainly expressed at the injection site 12 h after injection, but both gene expression of C55 DNA and plasmids then mainly occurred at the injection site from 2 – 47 days.

3. From fig. 3c and onwards, the authors switch from reporting transfection efficiency to reporting fluorescence intensity. I think the authors should continue to report transfection efficiency (fraction of fluorescent cells) for all the samples. The value of the mean fluorescence per cell has a quite broad distribution, as can be seen in fig. 6d, and the value of comparing means can be questioned. The transfection efficiency, thus remain an important complement.

Response: We thank the reviewer for this suggestion. The corresponding transfection efficiencies have now been added to the revised supplementary materials (Figs. S13, 15, 17, 21, 24, 25, 30, 32-35), and discussed in the revised main text.

Other comments:

Line 110: Why was pl DNA cleaved?

Response: The product information (Pub. No. MAN0013722, Thermo Fisher) shows that S1 Nuclease degrades single-stranded nucleic acids, and S1 Nuclease also cleaves dsDNA at the single-stranded region caused by a nick, gap, mismatch or loop. So we think it is possible that some of the open circular DNA form of our plasmids were cleaved by S1 nuclease.

Line 119: Shouldn't this be referencing fig. 1e? Or is this another experiment? If so, I think it should be added to the comparison in 1e.

Response: Thank you very much for pointing out the mistake. Fig. 1e has been added to the correct location in the revised manuscript.

Line 124. Are there single-stranded DNA viruses? I.e. not bacteriophages, but mammalian single-stranded DNA viruses? If yes, those should be cited. If no, rephrase. (to my knowledge there aren't any, this work makes one assume there should be ssDNA viruses actually)

Response: Yes, there are. Porcine circovirus diseases are caused by single-stranded DNA viruses (Anim. Health Res. Rev. 2005, 6,119-42).

Line 166 and fig. 2. Some cell lines show significantly higher transformation efficiency for the plasmid form. Authors should comment on this as they have chosen the cell line with the highest efficiency which has about 20% higher efficiency than the second highest and is not a human cell line (MDCK is a canine cell line).

Response: Thank you for the comment. Gene-coding C_{ss} DNA has a relatively high expression level in some cell lines. There may be three reasons for this. 1, For some cell lines with high expression level of C_{ss} DNA, lip2000 can easily pass through their cell membrane and enter the cell. 2, Compared to plasmids, notably, in some of the cell lines (such as HELF and WRL-68) C_{ss} DNA resulted in higher expression levels than the plasmid, possibly due to the relatively lower molecular weight and higher flexibility of single-stranded DNA, which potentially makes it easier to compact and complex with transfection reagents, and potentially abundant DNA replication machinery (required for the conversion of C_{ss} DNA into double-stranded form) in these cells, eventually resulting in increased gene expression levels. 3, MDCK cells proliferate faster than most of mammalian cells and are more conducive to the entry of C_{ss} DNA into the nucleus, eventually resulting in high gene expression level.

Line 187. The C_{ss}-Luciferase shows markedly different biodistribution than the plasmid. The authors should comment on this.

Also: There are no methods sections for the animal experiments nor the lipid nanoparticle packaging. It is hard for me to judge this experiment without further explanation of what was done, but it appears this was a different delivery mechanism than what is described in the rest of the paper.

Response: We would like to thank the reviewer for this observation. As the reviewer pointed out, the C_{ss} Luciferase was mainly expressed in liver of mice, but the plasmid was mainly expressed at the injection site 12 h after injection. We have added some discussion in the revised main text to address this point: "We surmise that some of the LNPs would enter the liver for both C_{ss} DNA and plasmid DNA. However, since the plasmid Luciferase (5984 bp) has the bacterial backbone sequence (required for the production of plasmid DNA), such as an antibiotic resistance gene, an origin of replication, etc., the plasmid may be more easily degraded and cleared by the liver, leading to comparatively stronger luciferase expression from C_{ss} DNA within 12 h." To further illustrate the in vivo expression of C_{ss} DNA, we present the long-term expression of C_{ss} DNA after 0.5 – 47 days (see our reply to question #2 above).

In addition, the corresponding experimental details have been added to the methods section in the revised manuscript as follows: "In vivo bioluminescence imaging: Five BALB/c mice (6-week-old females) were segmented into three groups, one mouse served as the negative control group, and the rest were evenly divided into plasmid (positive control) and C_{ss} DNA groups. Mice in the experimental groups were injected intramuscularly with 40 µg of nucleic acid substrate (luciferase plasmid and C_{ss} DNA), which was encapsulated in LNPs suspended in 50 µL DPBS buffer (pH 7.4). And the negative control group was injected PBS buffer (50 µL). Bioluminescence imaging was performed using the In Vivo Imaging System Lumina (IVIS) Lumina III imaging system (PerkinElmer) after 0.5 – 47 days. All mouse experiments were approved by the Animal Care and Use Committee of Institute of Basic Medicine and Cancer (IBMC), Chinese Academy of Science(2022R0004)."

Fig. 2e, what does "after i.m." mean?

Response: "after i.m." means "after intramuscular injection". The definition is now given in the revised Fig. 2e.

Line 248 and fig. 3. It seems a bit arbitrary to use a pair of oligos spaced at the half length of the construct. Why not start from single oligos with strong enhancing or inhibitory effect and seeing if the addition of another oligo changes that? Or would there be a synergistic effect between two enhancing oligos or two suppressing oligos? If the authors could provide a rationale for the quite specific lengths chose, that would help the reader. It is also hard to see what conclusions can be drawn from the screening in e. and g.

Response: We thank the reviewer for this suggestion. In fact, to further explore the effect of blocking strands on the expression of C_{ss} DNA, and its relation to the underlying C_{ss} DNA sequence, we designed another set of 48 nt long blocking strands that cross-linked the C_{ss} DNA in a different way. We specifically targeted the CMV promoter, the EGFP coding region and the non-coding region of the C_{ss} DNA with blocking strands denoted CB, EB, and NB, respectively. In total, we tested 49 blocking strands for which we systematically varied the position and the distance between the cross-linking positions within the respective domains (in the revised Supplementary Fig. 18). For instance, CB(L1) is a 48 nt blocking strand that binds to 24 nt at the leftmost end of the CMV promoter and cross-links it with another 24 nt long domain in the promoter, creating a loop of 10 nt length in the C_{ss} DNA. In the same way, CB(L2), ... CB(L6) bind to the same leftmost domain, but connect them to sequences with increasing distance (and thus create larger loops up to 460 nt length). Correspondingly CB(M1) ... CB(M5) bind to the middle portion of the promoter, and CB(R1) ... CB(R5) bind to the rightmost part. Following the same scheme, crosslinker strands were also designed for the other regions, i.e., EB(L1) ... EB(L4), EB(M1) ... EB(M4), EB(R1) ... EB(R3), NB(L1) ... NB(L4), NB(M1) ... NB(M3), NB(R1) ... NB(R3). The corresponding hybridization products were transfected into cultured MDCK cells for 24 h, and we found that 35% of the tested blocking strands had a significant effect on C_{ss} DNA expression (below 60%, with the lowest at only 22%, Fig. 4d in the revised manuscript, Supplementary Fig. 19).

On the whole, there is no clear trend that related the mean fluorescence intensity for the constructs of C_{ss} DNA and the position of the blocking strands. However, most of fused ssDNAs (48 nt) have a more significant inhibition on C_{ss} DNA expression than simple combinations of two individual

complementary strands (24 nt). We speculate that the two distinct possible DNA crossover geometries formed by the longer blocking strands (48 nt) with the C_{ss} DNA inhibit the DNA replication machinery - required for the conversion of C_{ss} DNA into double-stranded form - to different degrees, resulting in the observed difference in gene expression levels. The above discussion has been modified in the revised manuscript to make the reader understand the content of the paper more clearly.

Fig. 3a. I think the estimated number of bases paired should be shown along the nanometers. A gel of the denatured constructs could even directly show the distribution.

Response: We thank the reviewer for the suggestion. As the reviewer suggested, we conducted the corresponding experiments. The corresponding DNA samples (10 μ L, 500 ng) were separately treated with 10 μ L of formamide (a DNA denaturation reagent), and the mixture was annealed at 95°C for 20 min, then we attempted to run an agarose gel of the denatured constructs. However, as shown in Figure 10, there are many constructs that are not fully denatured, resulting in the generation of very small amounts of linear single-stranded DNA whose bands are difficult to see in the gel.

Figure 10. An agarose gel of the denatured constructs.

Fig. 3b, should be plotted with a continuous y-axis.

Response: Thank you for your suggestion. The revised Fig. 3b has been modified to include a continuous y-axis.

Fig. 4. These results are quite interesting compared to recently published results.

Response: In our discussion section, we compared our work with other publications as follows: "Recent work has reported that gene-encoding DNA origami objects transfected into mammalian cell lines via electroporation would denature in the cell, resulting in expression of the gene encoded the single-stranded scaffold. In contrast, in the present work we used lipofection as nucleic acid delivery method to transfect C_{ss} DNA into mammalian cells. Through systematic screening experiments, we found that already a single strand hybridized to specific sites on the C_{ss} DNA can significantly reduce gene expression. Notably, with the addition of more staples expression from the C_{ss} DNA becomes increasingly more suppressed (Fig. 4f in the revised manuscript). We speculate that liposome-coated C_{ss} DNA complexes – unlike naked DNA structures delivered by electroporation – are not immediately denatured in the cellular milieu, resulting in a different fate of the C_{ss} DNA in the cell."

Fig. 4. The DNA-lines should be thicker, colors are hard to see. I don't get the point of the dots in green or red in a hexagonal pattern around the genes to the right of the arrows.

Response: We fully agree with the reviewer's suggestion. The lines of DNA have been bolded in Fig.4. A description has been added to the legend in Fig. 4d of the revised manuscript as follows: "The dots in green and red are the expression efficiencies of C_{ss} EGFP(+) and C_{ss} mCherry(+), respectively, affected by the corresponding blocking strand."

Line 331: The claim that longer blocking strands show decreased inhibition originates from their increased propensity to form secondary structures should be backed by a simulation, at least with e.g. Nupack.

Response: The secondary structures of the longer blocking strands (80 nt and 100 nt) are now shown in Figure 11, which were obtained by UNAFold.

Figure 11 (Corresponding to Fig. S28 in the revised Supplementary materials). The secondary structure prediction of EGFP-block-80 nt and EGFP-block-100 nt by UNAFold.

Line 421: "Extremely" seems like an exaggeration as for example block E1/input C is comparable to the correct block - input combinations.

Response: We fully agree with the reviewer's comment. The text was toned down to correct this description in the revised manuscript.

Fig. 5a. The lines are too thin, colors are hard to see and in particular distinguish the different colors for blocking/unblocking.

Response: Thank you for the suggestion. The lines in Fig. 5a have been modified to make them clearer for the reader.

Fig. 5c. The arrows are a bit misleading. It is as if the images were taken on the same sample, which they are not.

Response: We fully agree with the reviewer's comment. The arrows have been removed for clarity.

Line 483-485: Can't the oligonucleotides act as primers as well?

Response (see our reply to Referee 2 on the same question): Based on the biochemistry of DNA replication and DNA polymerization, there should be a primer as all known DNA polymerases start at the 3'OH of a double-stranded primer. The primers needed for the replication of C_{ss} DNA might be either RNA primers or DNA primers.

During genome replication, double-stranded DNA genomes have to be unwound to serve as templates for semi-conservative replication, starting from RNA primers generated by dedicated RNA polymerases (primases, e.g., the primase subunit of Pol alpha). Thus, one potential replication pathway is the production of RNA primers from the C_{ss} DNA template (potentially in region that contains double-stranded secondary structure), and these RNA primers would then initiate replication of the C_{ss} DNA based on these RNA primers.

Another possibility – as rightfully suggested by this reviewer - is that DNA single strands binding to the C_{ss} DNA might act as primers themselves. As shown in Fig. 3e in the revised manuscript, we found that the addition of double-stranded regions to a C_{ss} DNA vector can either activate or inhibit gene expression, depending on the position. We therefore hypothesize that short complementary DNA fragments at specific sites can act as DNA primers, recruiting the appropriate DNA polymerases to convert the C_{ss} DNA into double-stranded form and therefore promote gene expression. Other complementary DNA fragments at certain sites apparently do not act as primers and rather act as blocking strands that can interfere with the replication process. We are still in the process of investigating the expression mechanism of C_{ss} DNA in greater detail.

Line 494. I think this could be toned down to simply stating that protein expression can be done in-vivo using LNP delivery and C_{ss}DNA. The data is from a single mouse experiment. (Which is also not properly explained, see above)

Response: We fully agree with the reviewer's suggestion. The biological repeat of the in vivo transfection was added to the revised supplementary materials (Fig. S9). In addition, the corresponding text has been modified in the discussion in the revised manuscript.

REVIEWERS' COMMENTS

Reviewer #1 (Remarks to the Author):

The authors correctly revised their manuscript. I do not have any further comment.

Reviewer #2 (Remarks to the Author):

Overall, the authors have well addressed previous comments. Though some questions are yet fully explained and could be answered by further experiments, these questions by large do not significantly influence the conclusion.

Reviewer #3 (Remarks to the Author):

The authors have addressed most of the issues that were pointed out and performed the necessary experiments with some exceptions.

The main issue is however that the results presented in Figure 8. of the rebuttal document show that the state of their CSS system cannot be altered after the transfection by the separately added trigger strands. The results of this experiments should definitely be included in the manuscript and discussed. Additionally, this seems to mean that the system is limited to its state that you set before the transfection, meaning that regulation (i.e. the ability to change its state after transfection) is not currently possible. So the authors should modify all parts of the text that state the opposite and be candid with this shortcoming in the main text.

In my view the manuscript still can be of interest with the results presented in this manner, and if the authors make the mentioned changes, it should be accepted to be published in the journal.

Response to Reviewer 1

The authors correctly revised their manuscript. I do not have any further comment.

Response: We appreciate the reviewer's positive assessment of the revised manuscript and thank you for the constructive comments.

Response to Reviewer 2

Overall, the authors have well addressed previous comments. Though some questions are yet fully explained and could be answered by further experiments, these questions by large do not significantly influence the conclusion.

Response: Thanks a lot, and we are indeed keep going to perform more experiments and hope to reveal more interesting phenomena and mechanism.

Response to Reviewer 3

The authors have addressed most of the issues that were pointed out and performed the necessary experiments with some exceptions. The main issue is however that the results presented in Figure 8. of the rebuttal document show that the state of their CSS system cannot be altered after the transfection by the separately added trigger strands. The results of this experiments should definitely be included in the manuscript and discussed. Additionally, this seems to mean that the system is limited to its state that you set before the transfection, meaning that regulation (i.e. the ability to change its state after transfection) is not currently possible. So the authors should modify all parts of the text that state the opposite and be candid with this shortcoming in the main text.

Response: Thank you very much for the suggestion. We have added the results of this experiments to revised Supplementary materials (Supplementary Figure 36) and more discussions in the revised main text. The corresponding revised parts with highlight are shown in the main text.

In my view the manuscript still can be of interest with the results presented in this manner, and if the authors make the mentioned changes, it should be accepted to be published in the journal.

Response: We appreciate the reviewer's kindness and suggestions.